# On the mechanisms of lysis triggered by perturbations of bacterial cell wall biosynthesis

Yoshikazu Kawai [1,5] ✉, Maki Kawai[1,5], Eilidh Sohini Mackenzie[2], Yousef Dashti [3], Bernhard Kepplinger [4], Kevin John Waldron [2,6] & Jeff Errington [1,5] ✉

Inhibition of bacterial cell wall synthesis by antibiotics such as β-lactams is thought to cause explosive lysis through loss of cell wall integrity. However, recent studies on a wide range of bacteria have suggested that these antibiotics also perturb central carbon metabolism, contributing to death via oxidative damage. Here, we genetically dissect this connection in *Bacillus subtilis* perturbed for cell wall synthesis, and identify key enzymatic steps in upstream and downstream pathways that stimulate the generation of reactive oxygen species through cellular respiration. Our results also reveal the critical role of iron homeostasis for the oxidative damage-mediated lethal effects. We show that protection of cells from oxygen radicals via a recently discovered siderophore-like compound uncouples changes in cell morphology normally associated with cell death, from lysis as usually judged by a phase pale microscopic appearance. Phase paling appears to be closely associated with lipid peroxidation.

Peptidoglycan (PG) cell wall maintains characteristic shapes of bacteria and protects the cells from fluctuations in internal osmotic pressure. Expansion of the cell during growth requires the insertion of new PG by the action of glycosyltransferase (GTase) and transpeptidase (TPase) enzymes[1,2]. In most rod-shaped bacteria, cell elongation is governed indispensably by the "Rod system", which involves a GTase called RodA[3–5] and one or more TPases belonging to the family of class B penicillin-binding proteins (bPBPs)[6,7]. The Rod system is regulated spatially and temporally to achieve orderly cell extension by cytoskeletal proteins of the MreB (actin-like) family, working together with accessory proteins MreC and MreD[2,8]. Some organisms have more than one MreB paralogue: in *Bacillus subtilis* there are three, and MreB, Mbl (MreB-like) and MreBH, are essential for normal cell elongation[9,10]. Many bacteria have a second PG synthetic system based on bifunctional class A PBPs (aPBPs) that have both GTase and TPase activities[1,11].

This system seems to insert new PG in a dispersed manner leading to growth in a spherical form[3], but its precise function is not clear, and all four aPBPs in *B. subtilis* can be deleted with only mild effects on cell growth[4,12].

β-Lactams are one of the oldest and still most widely used clinical antibiotics. They prevent the insertion of new PG by binding to PBPs and irreversibly inactivating their TPase activity[13]. It is generally considered that this ultimately causes explosive cell lysis by loss of cell wall integrity. Hence, other essential factors of the Rod system (e.g. MreB) also have the potential as targets for new antibacterial agents[14,15]. Nevertheless, in *B. subtilis*, various suppressor mutations have been shown to rescue the growth of *mreB* and/or *mbl* mutants[16,17]. Some of the suppressor mutations lay in genes directly connected with cell wall synthesis. However, many of the others targeted carbon and amino acid metabolism, highlighting that the bacterial response to inhibition

[1]Centre for Bacterial Cell Biology, Biosciences Institute, Faculty of Medical Sciences, Newcastle University, Richardson Road, Newcastle upon Tyne NE2 4AX, UK. [2]Bioscience Institute, Faculty of Medical Sciences, Newcastle University, Newcastle upon Tyne NE2 4HH, UK. [3]Faculty of Medicine and Health, University of Sydney, Sydney, NSW 2006, Australia. [4]Department of Molecular Microbiology, Faculty of Biotechnology, University of Wrocław, 50-383 Wrocław, Poland. [5]Present address: Faculty of Medicine and Health, University of Sydney, Sydney, NSW 2006, Australia. [6]Present address: Institute of Biochemistry and Biophysics, Polish Academy of Sciences, Warsaw 02-106, Poland. ✉e-mail: yoshikazu.kawai@sydney.edu.au; jeffery.errington@sydney.edu.au

of wall synthesis that results in cell death is complex. Curiously, the growth deficiency of *mreB*, *mbl* and other elongation mutants, including *rodA*, can be rescued by addition of high concentrations of magnesium ions ($Mg^{2+}$) to standard culture media[16,18–20], for reasons not fully understood.

The cell lysis caused by loss of cell wall integrity in the presence of cell wall active antibiotics, or lytic enzymes such as lysozyme, can be prevented by use of osmoprotective (isotonic) media[21–23]. However, these cell wall-defective cells normally do not grow and they still die. Recent work examining the lethal effect in *B. subtilis* and other Gram-positive bacteria linked the non-lytic death during cell wall inhibition to an increase in glycolytic flux, resulting in the production of reactive oxygen species (ROS) from the respiratory chain (RC)[24,25]. Crucially, physiological compensation for the metabolic imbalance, by reducing glycolytic activity or the RC pathway, counteracts the toxicity[24,25], and enables cell wall-independent proliferation in a state known as the L-form[26,27].

This is consistent with previous work discovering toxic perturbations in the tricarboxylic acid (TCA) cycle and RC activity in response to β-lactam treatment in *Escherichia coli*[28–30]. It has been shown that inhibition of *E. coli* PBP2 (a cell elongation-specific bPBP) by the β-lactam mecillinam triggers a futile cycle of cell wall synthesis and degradation that contributes to the lethal activity[31]. Recent follow-up work using metabolomics led to a model that the lethality from PBP2 inhibition is a specific consequence of toxic metabolic shifts induced by energy demand from multiple catabolic and anabolic processes triggered by the PG futile cycling[32]. The metabolic shifts include central carbon oxidation pathways and ATP utilization, ultimately leading to a dysregulated cellular redox environment. Similarly, the lethality to *Mycobacterium bovis* of cell wall-active antibiotics depends on an increase in ATP production[33]. An increase in ROS production caused by β-lactams has been reported in a wide range of bacteria, including *Pseudomonas* species[34], *Enterococcus faecalis*[35] and *Vibrio cholerae*[36]. ROS-dependent cell death in L-forms is also evident in *E. coli* and *Streptobacillus moniliformis*[25,37–39]. These results highlight that dysregulation of cell wall metabolism in the target-proximal triggers toxic consequences from downstream metabolic pathways that contribute to bacterial cell death.

We recently identified and purified an actinomycete natural product compound called mirubactin C (MC) that rescues the growth of *B. subtilis mreB* and *mbl* mutants[40]. MC is a derivative of a known siderophore, mirubactin A (MA)[41,42]. Siderophores are iron-chelating metabolites that are synthesised and secreted by microorganisms to solubilize $Fe^{3+}$ for its uptake in aerobic environments[43]. MC also binds iron and results in reduced cellular iron content in *B. subtilis*, consistent with a model in which it acts as an extracellular iron chelator that restricts iron bioavailability[40]. However, it remains unclear how iron utilization may have a role in the lethal effect of the cell wall mutants. Curiously, the growth-rescue effect is not induced by its larger relative MA[40].

Here, we dissect the physiological connections between upstream perturbations of cell wall inhibition and downstream metabolic effects that contribute to ROS-mediated lethality of cell wall mutations in *B. subtilis*. We show that MC sequesters iron and prevents its uptake, thereby counteracting oxidative damage via lipid peroxidation (LPO), likely by avoiding the prooxidant effects of redox-active iron[44]. Remarkably, we found that when mutants are rescued by MC, morphological abnormalities such as bulging and twisting that are generally assumed to be a precursor to cell death by lysis, are largely unaffected. The main effect is on loss of the "pale" cellular appearance, often associated with the term lysis, which is presumably due to leakage of cell contents and reduced cytoplasmic density. We show that the pale cellular appearance is likely due to the perturbation of membrane integrity by LPO. These results provide insights into the mechanisms of bacterial cell death and antibiotic action. We also provide a view of the differentiated roles of the widely conserved Rod and aPBP mechanisms of bacterial cell wall synthesis.

## Results

### Mirubactins as probes for iron uptake and utilization in *B. subtilis*

Our previous work showed that treatment of *B. subtilis* cells with MC substantially reduces cellular iron content (about 25% iron of untreated cells) and rescues the growth of both *mbl* and *mreB* mutants in complex-rich media[40]. The decreased cellular iron content suggested that MC may act extracellularly by sequestering iron (Fig. 1a). To understand the mechanism underlying this iron starvation, we cultured *B. subtilis* wild-type 168CA and various mutants impaired in known iron transporters (Fig. 1b)[45] under iron-limiting conditions (Spizizen minimal medium; SMM)[46], with glucose and ammonium as carbon and nitrogen sources, but without added iron. (Note that *B. subtilis* 168CA and its derivatives are unable to synthesize the siderophore bacillibactin due to the presence of the *sfp*[0] mutation[45,47].) All of the strains grew well on a normal complex medium, i.e. iron-replete nutrient agar (NA, estimated to contain 0.274 mg/l Fe[48]), without obvious impairment (Supplementary Fig. 1a, no add). In the presence of iron-free synthetic MC, they showed a mild slow-growth phenotype (Supplementary Fig. 1a, +MC), as seen previously[40]. On iron-limiting SMM plates, growth occurred in each of the strains except the *fecC* mutant (Fig. 1c, no add), lacking the ferric citrate transporter (Fig. 1b), consistent with available iron (as a contaminant from the added chemicals and/or leached from glassware) being primarily complexed with citrate in this medium. In the presence of MC, growth was strongly inhibited in all of the tested *B. subtilis* strains (Fig. 1c, +MC). This inhibitory effect was largely alleviated by addition of ferric citrate to the medium, except for the ferric citrate transporter mutant (Δ*fecC*) (Fig. 1c, +MC +Fe³⁺Cit). These results were consistent with MC causing iron deficiency, likely by sequestering iron, thereby preventing its uptake and/or utilization (Fig. 1d).

In contrast, in the presence of MA (a natural compound purified in its iron-free form[40]), the growth deficiency of the *fecC* mutant (Δ*fecC*) was ameliorated, whereas the growth of cells lacking the Fe-siderophore transporter (Δ*feuA*) was impaired (Fig. 1c, +MA). This suggests that MA can be used as a source of iron for supporting *B. subtilis* growth and that uptake occurs as an Fe-MA complex via the Feu transporter (Fig. 1d). However, adding higher amounts of MA inhibited *B. subtilis* growth (Supplementary Fig. 1b, c). Since MA is known to spontaneously decompose into MC[49], the growth inhibitory effect might be the consequence of an increased MC concentration under these conditions.

Consistent with the contrasting activities of MA and MC on iron utilization for the growth of *B. subtilis*, the growth deficiency caused by MC was counteracted in the presence of MA, provided that the Feu (iron-siderophore) transporter was functional (Fig. 1e).

Finally, we found that the Gram-negative bacterium *E. coli* did not show a growth defect in the presence of MC (Fig. 1f). This could be due to this organism being capable of taking up the Fe-MC complex as a source of iron. However, it seemed more likely that it was due to iron scavenging by the production of an endogenous *E. coli* siderophore, enterobactin, which is structurally similar to *B. subtilis* bacillibactin[50,51]. Consistent with this idea, the growth of wild-type *B. subtilis*, but not the *feuA* siderophore uptake mutant, was partially restored when grown adjacent to the *E. coli* strain (Fig. 1f). We assume that this growth restoration is due to *B. subtilis* taking up an iron complex of the enterobactin secreted by the *E. coli* strain. In support of this view, a bacillibactin-functional *B. subtilis* Marburg strain did not show a growth defect in the presence of MC (Fig. 1g). To conclude, it appears that MA acts as a classical iron siderophore, facilitating iron uptake via siderophore transporters, whereas MC also binds iron but is not taken up and can inhibit iron utilization.

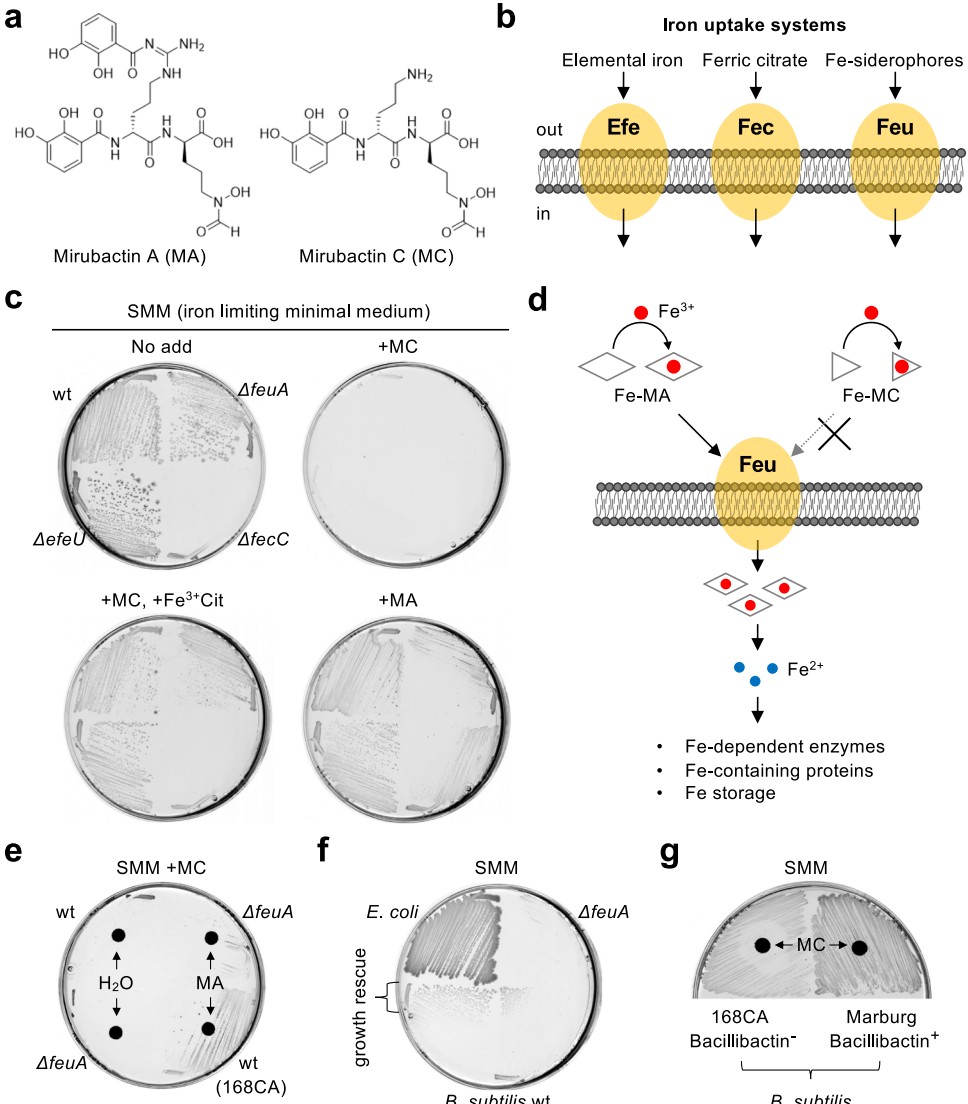

**Fig. 1 | Mirubactin C inhibits iron uptake and its utilization. a** Schematic representation of structures of Mirubactin A (MA) and C (MC). **b** Schematic representation of iron uptake systems in *B. subtilis*. **c** Growth inhibition by MC in iron-limiting minimal medium (SMM). *B. subtilis* strains 168CA (wild-type), YK2739 (*ΔefeU*), YK2740 (*ΔfecC*) and YK2741 (*ΔfeuA*) were streaked on SMM plates with or without 20 μg/ml MA, 10 μg/ml MC and/or 100 μM ferric-citrate (Fe³⁺Cit), and incubated for 30–42 h at 37 °C. **d** Schematic representation of effects of Mirubactins on iron uptake and utilization. **e** Uptake and utilization of iron in *B. subtilis* mediated by MA. 168CA (wild-type) and YK2741 (*ΔfeuA*) were streaked on a SMM plate in the presence of 10 μg/ml MC, with paper disc containing 6 μl of $H_2O$ or 20 mg/ml MA, and incubated for 30–42 h at 37 °C. **f** Uptake and utilization of iron in *B. subtilis* mediated by *E. coli*. *E. coli* strain BW25113 and *B. subtilis* strains (wild-type 168CA and *ΔfeuA*) were streaked on a SMM plate containing 10 μg/ml MC, and incubated for 30–42 h at 37 °C. **g** Growth of *B. subtilis* Marburg strain in the presence of MC. 168CA (bacillibactin⁻) and Marburg (bacillibactin⁺) strains were streaked on a SMM plate with paper disc containing 6 μl of 5 mg/ml MC, and incubated for 30–42 h at 37 °C. The figures are representative of at least three independent experiments.

## MC rescues *mbl* mutant cell death but not morphological change

In light of the above experiments, it was interesting to revisit the question of how MC (and $Mg^{2+}$) rescue the growth of an *mbl* mutant (Fig. 2a). In liquid nutrient broth (NB) with added $Mg^{2+}$, *mbl* mutant cells grew well, although they were slightly shorter and wider than wild-type cells (Supplementary Fig. 2a), as described previously[16]. When the culture was diluted into fresh NB without added $Mg^{2+}$, the cells took on abnormal bulging and twisted morphologies, and many cells became phase pale (Fig. 2b and Supplementary Fig. 2a). It is generally assumed that the phase pale or "ghost" appearance of cells treated with cell wall inhibitors is due to leakage of cell components through a damaged cell wall[1,22,52,53]. However, to our surprise, in NB without added $Mg^{2+}$ but with MC, although the abnormal twisted-shape was often observed, growth was sustained without the

appearance of significant numbers of phase pale cells (Fig. 2b and Supplementary Fig. 2a). It thus appears that the *mbl* mutant suffers from two distinct morphological abnormalities: shape changes, including bulging and twisting; and the phase pale appearance that may be the main signature of cell death. Furthermore, MC uncouples these effects and specifically rescues the phase pale (lysis) phenotype.

## *mbl* mutant cell death is dependent on iron utilization

The growth rescue effect of MC was abolished by the addition of ferric chloride ($FeCl_3$) or ferric citrate (Fe³⁺Cit) to the culture medium (Fig. 2c, panels 1 and 2), suggesting that rescue was dependent on reduced iron availability or uptake. To test this we used a general iron chelator, citrate, and found indeed that this rescued *mbl* mutant growth (Fig. 2c, panel 3). This growth rescue was abolished by further addition of ferric chloride (Fig. 2c, panel 4). In addition, when available

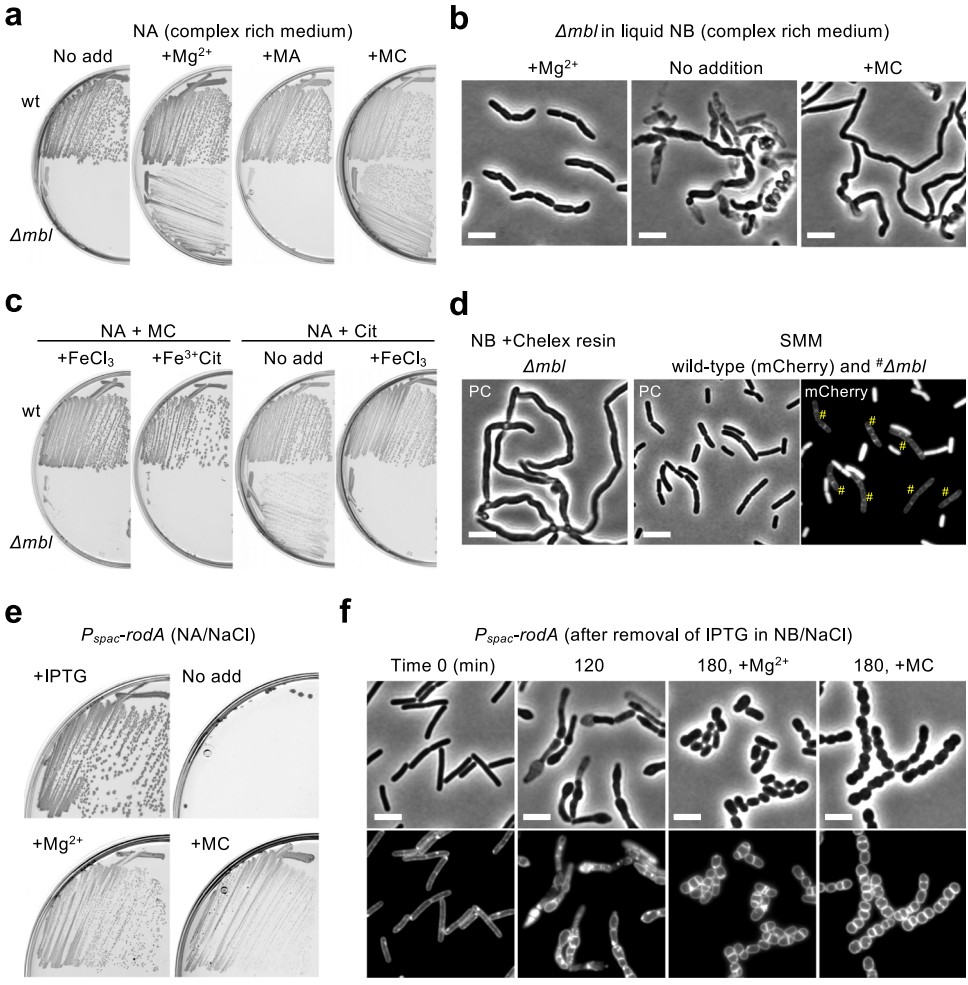

**Fig. 2 | Growth rescue of elongation mutants by restricting iron availability.**
**a** Growth rescue of an *mbl* mutant by MC. *B. subtilis* strains 168CA (wild-type) and
YK2638 (*Δmbl*) were streaked on NA plates with or without 10 mM MgSO₄ (Mg²⁺),
10 μg/ml MA or MC, and incubated for 18 h at 37 °C. **b** Suppression of phase pale
effect in an *mbl* mutant by MC. Phase contrast micrographs of exponentially
growing *mbl* mutant cells in liquid NB containing 10 mM Mg²⁺. The cells were
diluted into fresh NB with or without 10 μg/ml MC and incubated for 2–3 h. **c** Effects
of iron availability on *mbl* mutant growth in complex rich medium. Wild-type and
*Δmbl* strains were streaked on NA plates containing 10 μg/ml MC or 5 mM citrate
(Cit), with or without 50 μM ferric chloride (FeCl₃) or ferric citrate (Fe³⁺Cit) and
incubated for 18 h at 37 °C. **d** Effects of iron availability on cell morphology in an
*mbl* mutant. YK2265 (wild-type expressing mCherry) and YK2638 (*Δmbl*) strains
were cultured in NB with added 10 mM Mg²⁺ at 37 °C. *Δmbl* cells were diluted into
fresh NB pre-treated in the presence of 1% Chelex resin (no added Mg²⁺), and phase

contrast (PC) micrographs were captured after 90 min incubation (left panel).
Mixture of *Δmbl* and wild-type cells (expressing mCherry) were diluted into SMM,
and phase contrast and the corresponding fluorescent micrographs were captured
after 180 min incubation. **e** Growth rescue of a *rodA* mutant by MC. YK2245 (*P_spac*-
*rodA*) was streaked on NA plates containing 100 mM NaCl (for osmoprotection),
with or without 0.1 mM IPTG, 10 mM Mg²⁺ or 40 μg/ml MC, and incubated for 24 h
at 37 °C. **f** Suppression of phase pale effect in a *rodA* mutant by MC. Phase contrast
and the corresponding membrane staining images of exponentially growing *P_spac*-
*rodA* strain (YK2245) in liquid NB containing 0.1 mM IPTG (Time 0). The cells were
diluted into fresh NB with or without 10 mM Mg²⁺ or 10 μg/ml MC. Phase contrast
and the corresponding membrane staining images of the cells were captured after
2–3 h incubation as indicated. Scale bars represent 5 μm. The figures are repre-
sentative of at least three independent experiments.

iron in NB was reduced by pre-treatment with Chelex resin, *mbl* mutant
growth was sustained with the expected abnormal morphology but
without the appearance of significant numbers of phase pale cells
(Fig. 2d), just as was seen for MC treatment (Fig. 2b). Thus, the lethal
effect of *mbl* mutation under normal culture conditions is dependent
on the availability or utilization of iron.

Consistent with this, *mbl* mutant cells slowly but significantly grew
on iron-limiting SMM plates supplemented with 4 mM Mg²⁺ (the lowest
amount of Mg²⁺ required for robust growth of wild-type cells under
these conditions) (Supplementary Fig. 1d, e). Increasing Mg²⁺ to 10 mM
only slightly improved *mbl* mutant growth (Supplementary Fig. 1d,
panel 2). The addition of MC abolished growth of both wild-type and
*mbl* mutant strains (Supplementary Fig. 1d, panel 4), as described above
(Fig. 1c), whereas the addition of MA slightly reduced *mbl* mutant
growth likely by acting as a source of intracellular iron (Supplementary

Fig. 1d, panel 3). However, the addition of ferric chloride improved *mbl*
mutant growth (Supplementary Fig. 1e). We observed the cell mor-
phology of an *mbl* mutant in liquid SMM (without added iron), and
found that growth was sustained with no detectable morphological
defects (Fig. 2d). Thus, factors other than iron could also contribute to
the *mbl* rescue phenotype under these conditions.

## Mirubactin C rescues growth of other elongation mutants
We previously showed that MC also rescues the growth of an *mreB*
mutant, affected in the Rod elongation pathway, but not that of a
mutant lacking aPBP activity – the alternative cell wall expansion sys-
tem ("*Δ4* mutant")[40]. We hypothesized that MC would rescue mutants
affected in other components of the Rod system (Supplementary
Fig. 2b). Repression of *rodA* (using an IPTG-dependent promoter, *P_spac*)
abolished growth on NA plates containing (osmoprotective) NaCl but,

as expected, growth was rescued by addition of 10 mM Mg$^{2+}$ (Fig. 2e)[20]. Importantly, growth was also rescued by addition of MC, without added Mg$^{2+}$ (Fig. 2e). In liquid NB, as for the *mbl* mutant, *rodA* repression resulted in bulging and cell death (phase pale effect) (Fig. 2f). However, in the presence of added Mg$^{2+}$ or MC, the cells remained phase dark, while switching to a spherical form of growth (Fig. 2f). Thus, cell death upon loss of RodA function is also probably associated with altered iron uptake or utilization, and MC uncouples this iron-mediated cell death from the morphological abnormalities, in this case a rounded shape, just as for the *mbl* mutant (Fig. 2b). The growth rescue effect, produced either by added Mg$^{2+}$ or MC, was also observed in a strain lacking the *mreBCD* genes (Supplementary Fig. 2c).

We confirmed that the continued growth of *rodA*-depleted cells supported by MC is dependent on the aPBP system (Supplementary Fig. 2d), as expected based on the complementary functions of the Rod and aPBP systems during cell wall expansion[3,4].

## Mirubactin C prevents cell lysis during PG precursor depletion

The functioning of the Rod system is dependent on the supply of PG precursors. We therefore tested the effects of MC on cell growth during the depletion of PG precursors by using a $P_{spac}$-fused *murGB* construct. MurG protein catalyses the final step in lipid II synthesis, while MurB acts earlier in the precursor pathway[2] (see below). When growing cultures (IPTG supplemented) were diluted into fresh NB without IPTG, the cells stopped growing and lysed within 2 h after *murGB* repression (Fig. 3a). In the presence of MC, the cells again stopped growing but the cell lysis was clearly alleviated (Fig. 3a). Many cells had taken on a spherical form after 3 h of *murGB* repression (Fig. 3b, c, and Supplementary Fig. 3a). A similar rescue effect by MC for growth with reduced *murGB* expression was observed on solid NA plates (Supplementary Fig. 3b). Thus, MC can rescue the lethal effects of inhibiting not only Rod-complex function but also the supply of PG precursors, by uncoupling the cell death from the abonormal bulging. We have previously demonstrated that the morphological changes during depletion of PG precursors is due to dispersed PG synthesis by the aPBP system[22]. The switch to spherical growth upon *murGB* repression in the presence of MC suggests that the aPBP system is able to operate not only when the Rod system is perturbed, but also when precursor availability is reduced.

In the presence of added Mg$^{2+}$, growth significantly continued for a while after *murGB* repression, and then the cells stopped growing and started to lyse (Fig. 3a, b). In this case, spherical cells were less evident (Fig. 3b and Supplementary Fig. 3a), suggesting that the growth rescue occurs by a mechanism distinct from that of MC, as previously discussed[40].

## *mbl* mutant lethality is rescued by overexpression of the *murG* gene

In the course of carrying out the $P_{spac}$-*murGB* experiments, we titrated the levels of IPTG needed for the growth of otherwise wild-type and *mbl* mutant strains. The *mbl*+ strain showed strong growth down to 0.04 mM IPTG (Fig. 3d). At this concentration of IPTG, the *mbl* mutant failed to grow but, as expected, growth was rescued by the addition of Mg$^{2+}$ or MC (Fig. 3d, right panels), so presumably 0.04 mM IPTG provides sufficient PG precursor synthesis for growth in both wild-type and *mbl* mutant cells. Unexpectedly, however, at 1 mM IPTG, the *mbl* mutant was able to grow without added Mg$^{2+}$ or MC. In liquid NB with added Mg$^{2+}$, the *mbl* mutant grew, with a typical rod-shaped morphology, in the presence of either 0.05 or 1 mM IPTG (Fig. 3e). When the cultures were diluted into fresh NB without added Mg$^{2+}$, the cells with 0.05 mM IPTG became phase pale but those with 1 mM IPTG did not (Fig. 3e). Thus, an increased level of *murGB* expression can overcome the lethal effects of an *mbl* mutation.

To test which of the genes, *murG* or *murB*, was responsible for the rescuing effect we constructed strains to overexpress each of them individually. As shown in Fig. 3f, rescue only required overexpression of *murG*. Our results suggest that disruption of the *mbl* gene may interfere with the function of MurG, plausibly by affecting the membrane localization or stability of the protein.

## *glmU* expression levels impact lethality of the *mbl* mutant

MurG catalyses the final step in lipid II synthesis by adding the nucleotide sugar UDP-GlcNAc to lipid I (Fig. 4a)[2]. The above rescue effect by *murG* overexpression could be due to decreased intracellular UDP-GlcNAc levels. To test this possibility we placed the *glmU* gene, required for UDP-GlcNAc synthesis (Fig. 4a)[54], under $P_{spac}$ control. Since *glmU* is essential for PG precursor synthesis, no growth occurred at low IPTG concentrations (at or below 0.05 mM) in either an *mbl* mutant or in its parental *mbl*+ strain (Fig. 4b, panel 1 and Supplementary Fig. 4a). A concentration of 0.1 mM provided sufficient expression to support at least partial growth of both the wild type and the *mbl* mutant strains (Fig. 4b, panel 2). However, whereas substantial growth occurred in the *mbl*+ background when more IPTG was added, to further increase *glmU* expression, growth of the *mbl* mutant was abolished above 0.5 mM IPTG (Fig. 4b, panel 3 and Supplementary Fig. 4a). Strikingly, this growth inhibition was again largely overcome in the presence of MC (Fig. 4b, panel 4). Thus, the *mbl* mutant does not appear to be able to tolerate levels of UDP-GlcNAc synthesis that support normal growth in wild-type cells and the toxicity effect is probably again dependent on iron availability.

At the minimal permissive concentration of IPTG (0.1 mM) for *glmU* expression, *mbl*+ cells had their normal rod-shaped phase dark appearance (Fig. 4c, d). The *mbl* mutant cells had a highly abnormal bulging and twisted appearance but notably the cells were virtually all phase dark, indicating a lack of cell lysis. The abnormal morphologies probably explained the poor growth on plates (Fig. 4b, panel 2). When *glmU* gene expression was increased (1 mM IPTG), the *mbl* mutant cells started to take on a more elongated form but they also began to undergo lysis (Fig. 4c, d), as shown above (Fig. 2b). Thus, it seems that the toxicity associated with UDP-GlcNAc synthesis can be separated from the morphological defects generated by the loss of the *mbl* gene.

## Sensitivity to Fosfomycin is also influenced by UDP-GlcNAc synthesis

Fosfomycin (FOS) is a broad-spectrum bactericidal antibiotic that prevents the first committed step in lipid I synthesis by inhibiting the MurA enzyme (Fig. 4a)[55]. Inhibition of MurA should also result in decreased utilization of UDP-GlcNAc for PG precursor synthesis, which, on the basis of the above experiments, might contribute to FOS toxicity. We took wild-type background cells bearing the $P_{spac}$-*glmU* construct and tested to see whether titration of expression (via IPTG concentration) would affect sensitivity to FOS. Remarkably, reducing *glmU* expression with 0.1 mM IPTG conferred much more resistance (smaller zone of growth inhibition) compared with the wild-type, and fosfomycin sensitivity returned to the wild-type level when *glmU* expression was restored by the addition of 1 mM IPTG (Fig. 4e). In contrast, *glmU* expression levels had no detected effect on sensitivity to vancomycin (Fig. 4e), which inhibits the insertion of new lipid II into the PG wall[56]. These results again support the hypothesis that a reduction of UDP-GlcNAc utilization contributes to the toxic effect when cell wall synthesis is perturbed.

## UDP-GlcNAc levels are affected by Mg$^{2+}$ but not MC treatment

We then directly measured the intracellular levels of UDP-GlcNAc in wild-type cells cultured with added Mg$^{2+}$ or MC. In the presence of Mg$^{2+}$, UDP-GlcNAc levels were substantially reduced (Supplementary Fig. 4b), consistent with a role for UDP-GlcNAc concentration in the toxic effects of cell wall perturbations. Nevertheless, addition of MC did not lead to reduced UDP-GlcNAc levels (Supplementary Fig. 4b). Similar results were obtained in *mbl* mutant cells (Supplementary

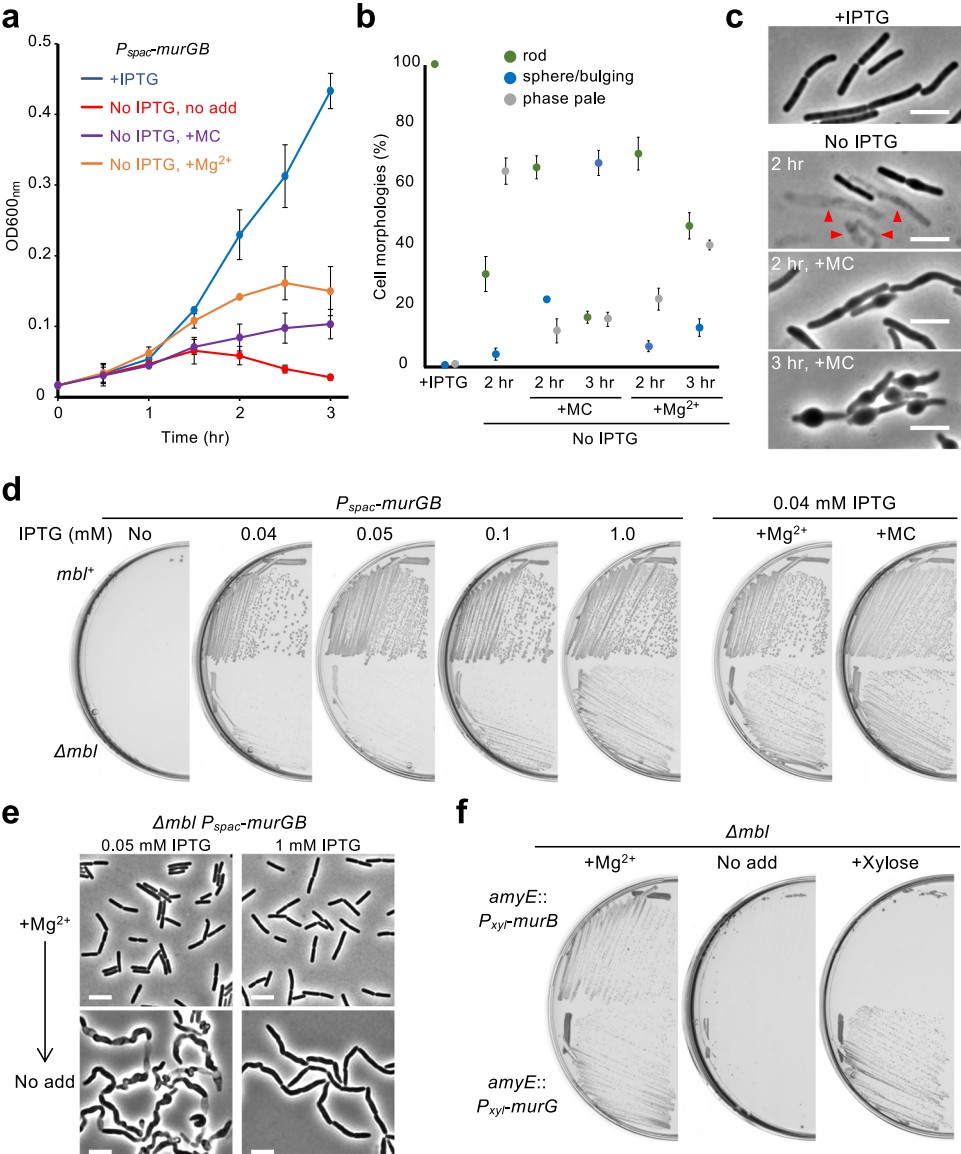

**Fig. 3 | Dysfunction of lipid II synthesis associates with an *mbl* mutant lethality.**
**a** Growth curves during *murGB* repression. *B. subtilis* strain YK1540 (*P_spac*-*murG*-*murB*) was cultured in NB containing 0.1 mM IPTG (blue line) at 37 °C. The exponentially growing cells were diluted into fresh NB (without IPTG, red line) with 10 μg/ml MC (purple line) or 10 mM Mg²⁺ (orange line) and cultured at 37 °C for OD₆₀₀ measurements. The means were obtained from three independent experiments. Error bars indicate standard deviation (SD). Source data are provided as a Source Data file. **b** Effects of MC on lethal effect and cell morphology during *murGB* repression. Cell morphologies of about 200 cells under each condition shown in panel a were classified into three types (rod, sphere/bulging or phase pale) based on the phase contrast images as shown in panel c and Suppremenratly a. The results were obtained from three independent experiments. Error bars indicate SD. Source data are provided as a Source Data file. **c** Phase contrast micrographs of *P_spac*-*murG*-*murB* (YK1540) cells were captured in the course of carrying out the growth curves

shown in panel a. Scale bars represent 5 μm. **d** Effects of *murGB* expression levels on *mbl* mutant growth. YK1540 (*P_spac*-*murG*-*murB*) and YK2665 (Δ*mbl* *P_spac*-*murG*-*murB*) strains were streaked on NA plates containing various concentrations of IPTG, with or without 10 μg/ml MC or 10 mM Mg²⁺. The plates were incubated for 18 h at 37 °C. **e** Effects of *murGB* expression levels on cell morphology of an *mbl* mutant. Phase contrast micrographs of strain YK2665 (Δ*mbl* *P_spac*-*murG*-*murB*) in liquid NB containing 10 mM Mg²⁺, with 0.05 mM or 1 mM IPTG (+Mg²⁺). The exponentially growing cells were diluted into fresh NB (no added Mg²⁺) with 0.05 or 1 mM IPTG and incubated for 2–3 h (No add). Scale bars represent 5 μm. **f** Growth rescue of an *mbl* mutant by *murG* overexpression. YK2700 (Δ*mbl* *amyE*::*P_xyl*- *murB*) and YK2701 (Δ*mbl* *amyE*::*P_xyl*- *murG*) were streaked on NA plates with or without 10 mM Mg²⁺ or 0.5% xylose, and incubated for 18 h at 37 °C. The figures are representative of at least three independent experiments.

Fig. 4c). Thus, the growth rescue by Mg²⁺ could occur via a reduction of intracellular UDP-GlcNAc levels, whereas MC probably acts by a mechanism downstream from that of Mg²⁺.

### Lethality of the *mbl* mutation involves glycolysis
GlmS acts as an anabolic-catabolic (PG wall - glycolysis) checkpoint by modulating the flux of a sugar-phosphate intermediate, fructose-6-phosphate (Fig. 5a, F6P)[57–61]. When the intracellular concentration of UDP-GlcNAc is high, GlmS activity is reduced. We previously showed

that blockage of the UDP-GlcNAc pathway by repression of *glmS* during cell wall inhibition results in the redirection of glucose metabolism towards glycolysis[25]. Importantly, the increased carbon flux through glycolysis acts as a serious impediment to the growth of cell wall-free L-forms. This correlates with an increased generation of toxic ROS from the RC pathway. Considering the above experiments, we wondered whether this might also be the case in *mbl* mutants. If so, slowing down glycolysis in an *mbl* mutant should rescue growth, as seen previously for L-forms[25]. We introduced *P_spac*-*gapA*, which encodes an essential

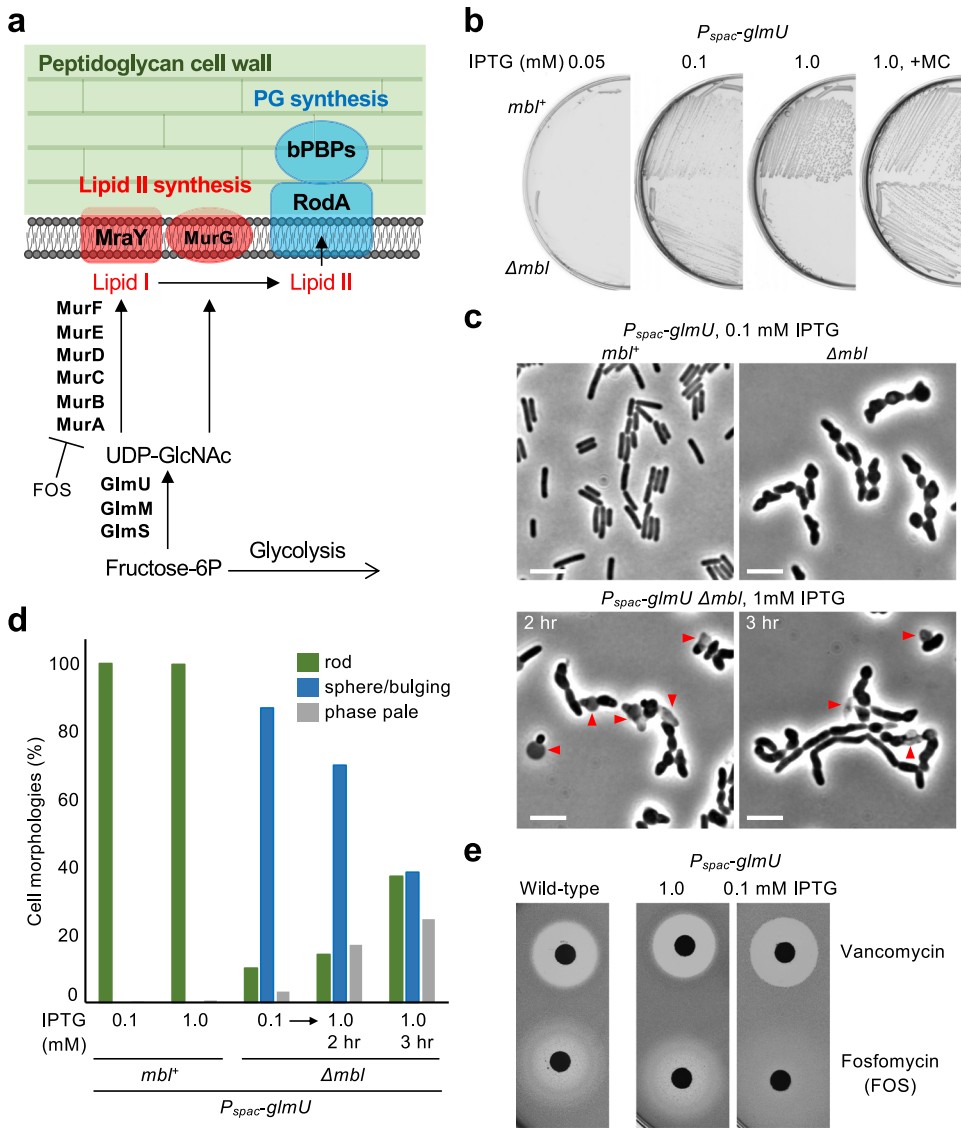

**Fig. 4 | UDP-GlcNAc synthesis associates with *mbl* mutant lethality. a** Schematic representation of PG precursor synthesis in *B. subtilis*. **b** Growth rescue of an *mbl* mutant by reducing UDP-GlcNAc synthesis. *B. subtilis* strains YK1538 ($P_{spac}$-*glmU*) and YK2687 (Δ*mbl* $P_{spac}$-*glmU*) were streaked on NA plates containing 0.1 or 1 mM IPTG, with or without 10 μg/ml MC, and incubated for 18 h at 37 °C. **c** Effects of *glmU* expression levels on cell morphology of an *mbl* mutant. Phase contrast micrographs of strains YK1538 ($P_{spac}$-*glmU*) and YK2687 (Δ*mbl* $P_{spac}$-*glmU*) in liquid NB containing 0.1 mM IPTG (upper panels). In the culture of YK2687, IPTG was added

to 1 mM, and phase contrast micrographs were captured after 2 and 3 h incubation (bottom panels). Scale bars represent 5 μm. **d** Cell morphologies shown in panel c were classified into three types (rod, sphere/bulging or phase pale). About 300 cells were examined for each condition. **e** Effects of *glmU* expression levels on fosfomycin sensitivity. Disc diffusion assay of *B. subtilis* strains (wild-type and $P_{spac}$-*glmU*) on NA plates with or without IPTG using paper discs with 6 μl of 1 mg/ml vancomycin or 50 mg/ml fosfomycin. The plates were incubated for 24 h at 37 °C. The figures are representative of at least three independent experiments.

glycolytic enzyme, NAD-dependent glyceraldehyde 3-phosphate dehydrogenase (Fig. 5a)[62], into an *mbl* mutant and titrated the levels of IPTG needed for growth. As shown in Fig. 5b, no growth occurred in either an *mbl* mutant or its isogenic *mbl*⁺ strain on NA plates without IPTG (panel 1), consistent with the expectation that the cells depend on generating energy through glycolysis under these conditions. At a low concentration of IPTG, just sufficient to allow growth of the wild-type (0.02 mM), the *mbl* mutant was able to grow (Fig. 5b, panel 2), whereas at higher IPTG concentrations (above 0.2 mM), growth was inhibited (Fig. 5b, panel 3 and Supplementary Fig. 5a). Just as in the experiments described earlier, growth inhibition was suppressed in the presence of MC (Fig. 5b, panel 4).

Consistent with the critical importance of glycolysis in the lethal effect, activating gluconeogenesis by supplying a gluconeogenic carbon source (malate) also rescued the growth of an *mbl* mutant (Supplementary Fig. 5b).

## NADH oxidation in the RC pathway associates with the lethality in *mbl* mutant

These results suggested that the utilization of sugar-phosphate intermediates through glycolysis may cause an increased generation of ROS from the RC pathway in *mbl* mutants. We previously found that several mutations promoting robust L-form growth work by reducing RC activity, which reduces the generation of toxic ROS[24]. The *hepS-menH-hepT* operon is one of the previously identified suppressor loci and the products are required for the biosynthesis of menaquinone (Fig. 5a), which is normally essential in electron transport and ATP generation in many Gram-positive bacteria[63]. We introduced a $P_{spac}$-*hepS* operon construct into an *mbl* mutant and tested the effects of reducing menaquinone synthesis on *mbl* mutant growth. As expected, the partial repression of menaquinone synthesis with 0.02 mM IPTG rescued the growth of the *mbl* mutant (Fig. 5c, panel 2). Growth was inhibited at higher levels of expression (IPTG >0.1 mM) and again the growth

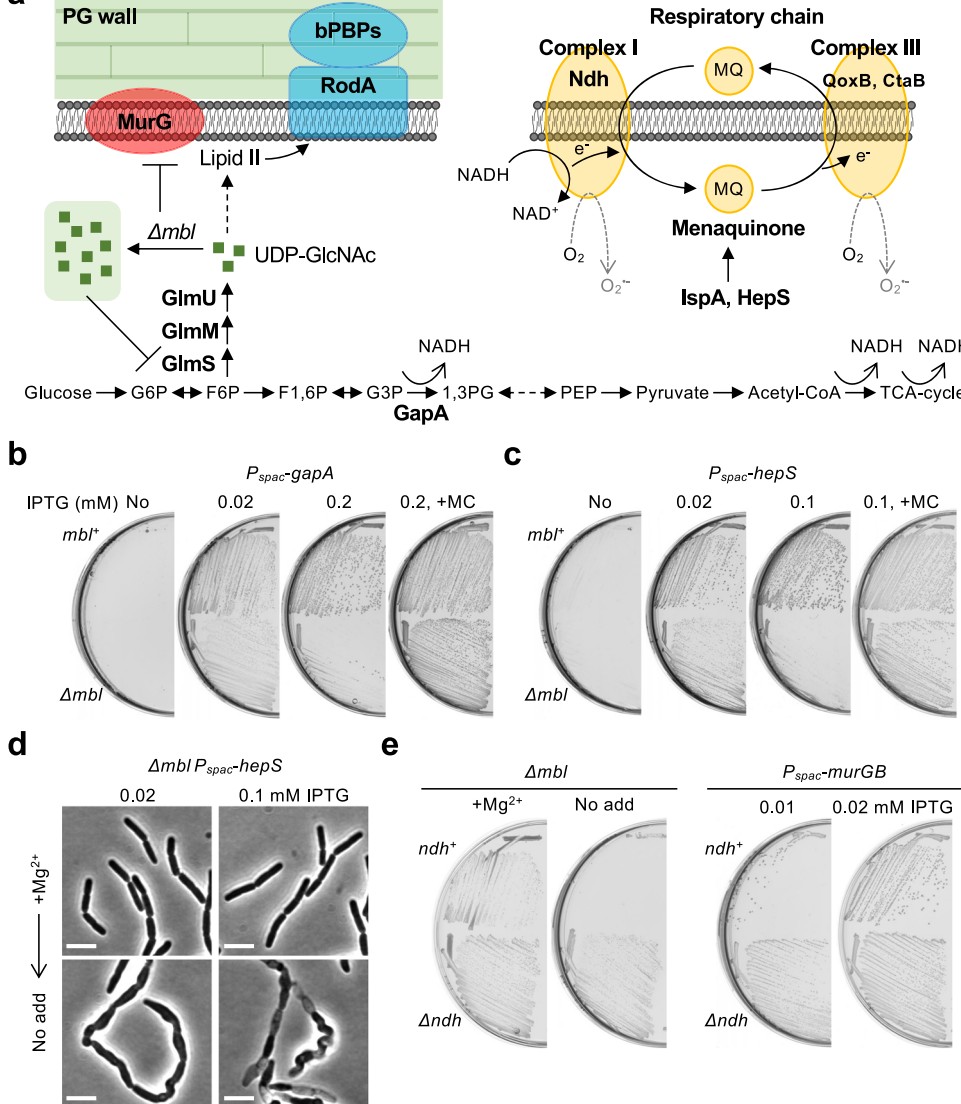

**Fig. 5 | Cellular respiration associates with *mbl* mutant lethality. a** Schematic representation of a link between PG precursor synthesis and central carbon metabolism in *B. subtilis*. **b** Effects of *gapA* expression levels on *mbl* mutant growth. YK1567 (*P_{spac}-gapA*) and YK2711 (*Δmbl P_{spac}-gapA*) were streaked on NA plates containing 0.02 or 0.2 mM IPTG, with or without 10 µg/ml MC, and incubated for 18 h at 37 °C. **c** Effects of menaquinone synthesis on *mbl* mutant growth. YK1450 (*P_{spac}-hepS-menH-hepT*) and YK2625 (*Δmbl P_{spac}-hepS-menH-hepT*) were streaked on NA plates containing 0.02 or 0.1 mM IPTG, with or without 10 µg/ml MC, and incubated for 18 h at 37 °C. **d** Effects of menaquenone synthesis on cell morphology of an *mbl* mutant. Phase contrast micrographs of a strain YK2625 (*Δmbl P_{spac}-hepS-*

*menH-hepT*) in NB containing 10 mM Mg²⁺, with 0.02 mM or 0.1 mM IPTG (+Mg²⁺). The exponentially growing cells were diluted into fresh NB (no added Mg²⁺) with IPTG and incubated for 2–3 h (No add). Scale bars represent 5 µm. **e** Growth rescue of an *mbl* mutant and *murGB* repression by disrupting NADH dehydrogenase. YK2638 (*Δmbl*) and YK2646 (*Δmbl Δndh*) were streaked on NA plates with or without 10 mM Mg²⁺ and incubated for 18 h at 37 °C (left panels). YK1540 (*P_{spac}-murG-murB*) and YK2689 (*Δndh P_{spac}-murG-murB*) were streaked on NA plates with 0.01 or 0.02 mM IPTG and incubated for 18 h at 37 °C (right panels). The figures are representative of at least three independent experiments.

inhibition was counteracted in the presence of MC (Fig. 5c, panels 3 and 4). In liquid NB with added Mg²⁺, the *mbl* mutant grew in a typical rod-shaped morphology in the presence of either 0.02 or 0.1 mM IPTG (Fig. 5d). When the cultures were diluted into the fresh NB but without added Mg²⁺, the cells with 0.1 mM IPTG became phase pale, but those with 0.02 mM IPTG did not (Fig. 5d).

We then tested the effects of various other L-form promoting mutations in the RC pathway (i.e. *ndh*, *ispA*, *qoxB* and *ctaB*[24]) to see whether they would also rescue *mbl* mutant growth. As shown in Fig. 5e and Supplementary Figure 6a, *ndh* (NADH dehydrogenase in complex I) and *ispA* (geranyltranferase for menaquinone synthesis) mutations rescued *mbl* mutant growth. However, no growth rescue occurred in the presence of *qoxB* (cytochrome quinol oxidase in complex III) or *ctaB* (heme O synthase in complex III) mutation. We also tested a

mutation in *sdhA*, which encodes succinate dehydrogenase forming complex II in the RC pathway[64], but again no growth rescue occurred (Supplementary Fig. 6a). Thus, electron transfer from NADH dehydrogenase to menaquinone in the NADH oxidation reaction, which generates superoxide as a by-product (Fig. 5a)[65], seems to associate with the *mbl* toxicity effect. We also tested the *ndh* mutation on the toxic effect of *murG* repression, and found that the growth deficiency at 0.01 mM IPTG was largely overcome (Fig. 5e, right panels), just as for treatment with MC (Supplementary Fig. 3b).

NADH is produced via various enzymes in central carbon metabolism (Fig. 5a), any or all of which could act as a source for toxicity in the *mbl* mutant. GapA activity in glycolysis is coupled to the reduction of NAD⁺ to NADH (Fig. 5a)[62] and reducing its activity can rescue the growth of the *mbl* mutant (Fig. 5b). We therefore tested mutations

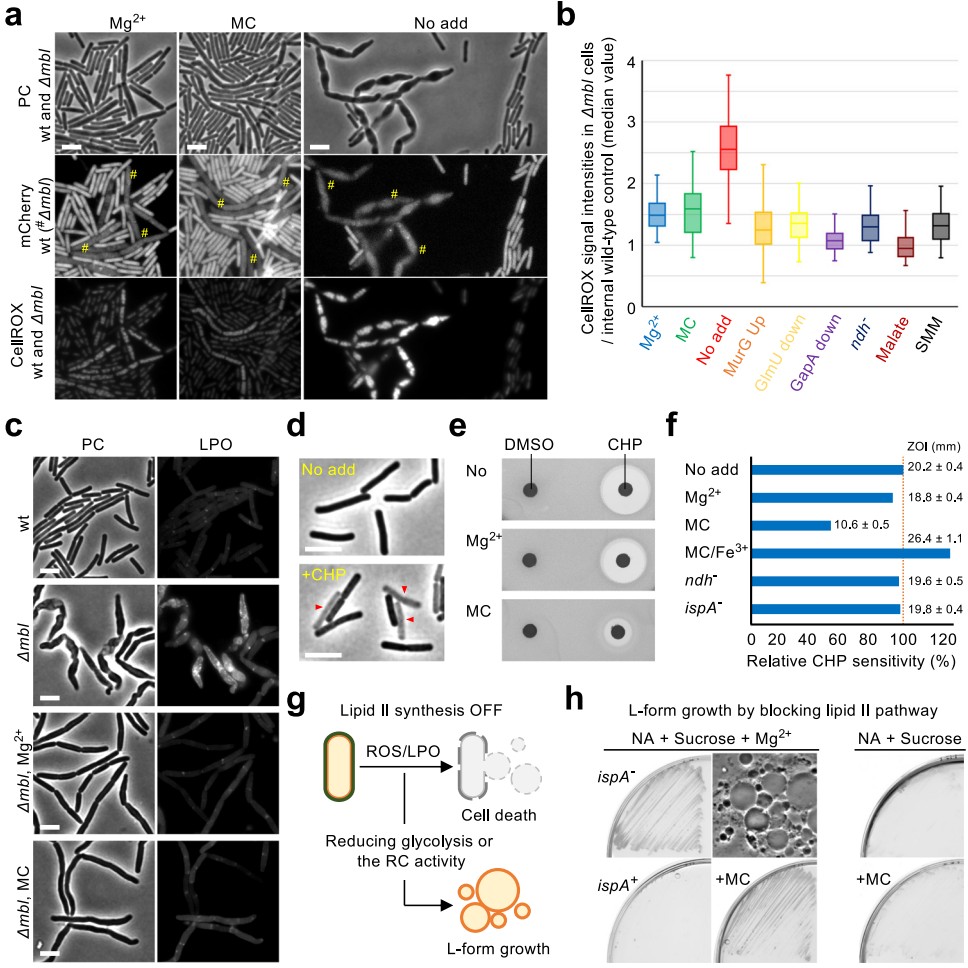

**Fig. 6 | MC treatment counteracts ROS-mediated cell death. a** ROS production in *mbl* mutant cells. YK2265 (wild-type expressing mCherry) and YK2638 (Δ*mbl*) were cultured in NB with added 10 mM Mg²⁺ at 37 °C, followed by CellROX Green treatment. Phase contrast (PC) and the corresponding fluorescent images (mCherry and CellROX) of mixtures of the wild-type and Δ*mbl* cells were captured (left panels). The cultures before CellROX treatment were diluted into fresh NB (no added Mg²⁺) with or without 10 µg/ml MC and incubated for 60 min, followed by CellROX treatment, before taking images. **b** YK2265 and *mbl* mutant strains (YK2638; Δ*mbl*, YK2701; Δ*mbl amyE::P_xyl*- *murG*, YK2687; Δ*mbl P_spac*-*glmU*, YK2711; Δ*mbl P_spac*-*gapA*, YK2646; Δ*mbl* Δ*ndh*) were cultured in NB containing 10 mM Mg²⁺ with appropriate supplements. The cultures were diluted in fresh SMM, or NB (no added Mg²⁺) with or without 10 µg/ml MC, 0.5% xylose (for YK2701), 0.1 mM IPTG (for YK2687), 0.05 mM IPTG (for YK2711) or 0.2% malate, and incubated for 60–90 min, followed by CellROX treatment, before taking images. The relative signal intensity of green fluorescence in *mbl* mutant cells (*n*=100) over internal wild-type control (mean value, *n*=100) were represented by boxplots. Boxplots represent the upper and lower quartile values (boxes), median (horizontal lines in the boxes) and most extreme data points within 1.5 times interquartile ranges (whiskers). Source data are provided as a Source Data file. **c** Lipid peroxidation (LPO) in *mbl* mutant cells.

Cultures of wild-type and *mbl* mutant cells in NB with 10 mM Mg²⁺ were diluted in fresh NB (no added Mg²⁺) with or without MC and incubated for 60 min, followed by C₁₁-BODIPY⁵⁸¹⁄⁵⁹¹ treatment, before taking images. **d** Effects of CHP treatment on cell morphology. Phase contrast micrographs of wild-type (168CA) cells in liquid NB before or after treatment with 10 mM CHP. **e** Disc diffusion assay on NA plates with or without 10 mM Mg²⁺ or 10 µg/ml MC using paper discs with 6 µl of DMSO or 250 mM CHP. The plates were incubated for 24 h at 37 °C. **f** Zones of growth inhibition (ZOI) were measured from four independent disc diffusion assays. The means and SD were shown. Wild-type (168CA), YK1714 (Δ*ndh*) and YK1395 (*ispA*⁻) was cultured on NA plates with or without added Mg²⁺, 10 MC and 50 µM ferric chloride (Fe³⁺). **g** Schematic representation of L-form generation and ROS-mediated killing during inhibition of lipid II synthesis. **h** Effects of MC and Mg²⁺ on L-form growth. LR2 (*ispA*⁻ *P_xyl*-*murE*) and BS115 (*P_xyl*-*murE*) strains were streaked on NA plates containing 0.5 M sucrose, with or without 20 mM MgSO₄ (Mg²⁺) and/or 10 µg/ml MC, and incubated for 2–3 days at 30 °C. The phase contrast micrograph of L-form cells was obtained from the left plate adjacent to the micrograph. Scale bars represent 5 µm. The figures are representative of at least three independent experiments.

affecting pyruvate dehydrogenase (*pdhA*), 2-oxoglutarate dehydrogenase (*odhA*) and malate dehydrogenase (*mdh*), and found that inactivation of *pdhA* or *odhA*, but not *mdh*, rescued growth of the *mbl* mutant (Supplementary Fig. 6b). Thus, the *mbl* toxicity effect seems to associate with production of NADH from central carbon metabolism.

### ROS-mediated cell death in the *mbl* mutant

The above results suggested that the toxic effects of *mbl* mutation are due to increased ROS production through the RC pathway. We therefore measured intracellular ROS levels using the ROS-sensitive permeable dye CellROX Green, which becomes fluorescent upon

binding DNA after being oxidized by superoxide (O₂⁻) or hydroxyl radical (•OH)[66]. We cultured wild-type and *mbl* mutant strains in NB with added Mg²⁺, and the exponentially growing cells were aerobically incubated with CellROX for 30 min. Figure 6a shows a mixture of the wild-type (expressing mCherry) and *mbl* mutant cells (left panels). An obvious CellROX fluorescence was detected in both strains, but the fluorescence was clearly weaker in wild-type cells than in *mbl* mutant cells (Fig. 6a, b, Mg²⁺). Cultures (before treatment with CellROX) were then diluted in fresh NB with or without MC and incubated for 60 min, followed by CellROX treatment. In the presence of MC, the pattern of CellROX fluorescence was similar to the case for added Mg²⁺ (Fig. 6a, b,

MC). Strikingly, in the absence of MC, strong fluorescence appeared in many *mbl* mutant cells but not in wild-type cells (Fig. 6a, b, No add). We confirmed that the growth rescue effects on *mbl* mutants under various conditions as described above (i.e. *murG* overexpression, *glmU* repression, *gapA* repression, *ndh* inactivation, addition of malate and SMM medium) all correlated with reduced ROS levels (Fig. 6b).

High levels of free radicals or ROS can damage cells by hitting a range of molecular targets[67,68]. Oxidative damage to lipids, lipid peroxidation (LPO), alters the physical properties of the cellular membrane, and generates a range of toxic effects in all cells[44], such as ferroptosis in eukaryotic cells[69]. It seemed possible that LPO could be the source of the phase pale, lytic effect seen in the various mutants described above. To test this, we took advantage of a fluorescent fatty acid analog, $C_{11}$-BODIPY[581/591] [24]. In a control experiment with wild-type cells, no clear fluorescence was detectable (Fig. 6c, wt). In an *mbl* mutant, obvious fluorescence was not observed in the presence of added $Mg^{2+}$ (Fig. 6c, $\Delta mbl$, $Mg^{2+}$), whereas the fluorescence became readily detectable when the added $Mg^{2+}$ was depleted (Fig. 6c, $\Delta mbl$). Crucially, the strong fluorescence in the *mbl* mutant was reduced in the presence of MC (Fig. 6c, $\Delta mbl$, MC). Thus, LPO seems to be a major cause for the phase pale effect in *mbl* mutants.

## Mirubactin C reduces lipid peroxidation by restricting iron availability

To investigate the mechanism underlying the suppression of LPO by MC, we made use of an exogenous inducer, cumene hydroperoxide (CHP), which is a stable organic oxidizing agent that acts to initiate and propagate endogenous ROS-independent LPO[44]. When *B. subtilis* wild-type cells were treated with CHP, a lethal phase pale effect was induced (Fig. 6d, red arrowheads), as seen for elongation mutants (Fig. 2b, f) or following the repression of lipid II synthesis (Fig. 3b, c). Importantly, the toxic effect of CHP treatment was significantly suppressed in the presence of MC, as indicated by a much smaller zone of growth inhibition (Fig. 6e, f). Redox-active transition metals, especially iron, can trigger exacerbating rounds of free radical-mediated LPO by CHP[44]. Crucially, the resistant effect of MC was counteracted in the presence of added ferric iron (Fig. 6f). Thus, MC prevents LPO-mediated toxicity likely by restricting iron bioavailability. In contrast, $Mg^{2+}$ did not confer significant resistance, and neither *ndh* nor *ispA* mutations influenced CHP toxicity (Fig. 6f).

SOD enzymes are an important antioxidant defense against superoxide[70]. They catalyse the dismutation of superoxide radical ($O_2^-$) (e.g. formed by electrons leaking from the RC) into molecular oxygen ($O_2$) and hydrogen peroxide ($H_2O_2$). The latter then acts as a source of hydroxyl radical ($\cdot OH$) formation via the Fenton/Haber-Weiss reaction cycle in the presence of redox-active iron[67,71,72]. We examined SOD activity in lysates of vegetative cells of wild type and *mbl* mutant cells in the presence of added $Mg^{2+}$ or MC. In the presence of MC, SOD activity was substantially elevated in the *mbl* mutant compared with the wild-type (Supplementary Fig. 7), consistent with increased production of superoxide from the RC pathway in the *mbl* mutant. Strikingly, addition of $Mg^{2+}$ did not lead to elevated SOD activity in either mutant or wild type cells (Supplementary Fig. 7). Thus, $Mg^{2+}$ could work to rescue *mbl* viability by preventing excess superoxide production, whereas MC treatment acts downstream to prevent the formation of hydroxyl radical and LPO by restricting iron bioavailability.

## Mirubactin C promotes L-form growth by avoiding ROS-dependent cell death

Our results suggest that iron starvation in the presence of MC works to rescue *mbl* mutant growth by preventing ROS-dependent cell death mediated through the RC pathway. If so, MC could also work to promote L-form growth by avoiding the LPO-dependent toxicity we previously reported (Fig. 6g)[24,25]. MC was not able to support the growth of

cells when *murGB* expression was completely inhibited on NA plates (Supplementary Fig. 3b). (Prolonged culture without the supply of PG precursors ultimately results in osmotic lysis due to the formation of cell wall lesions via ongoing autolytic activity under standard conditions[22]). To induce an L-form state, lipid II synthesis was blocked on isotonic NA plates supplemented with 20 mM $MgCl_2$ and 0.5 M sucrose[21]. As expected, based on our previous work, cells bearing an *ispA* mutation (inhibiting menaquinone synthesis) were able to switch into the L-form state and grow, whereas *ispA*[+] cells were not viable (Fig. 6h). Strikingly, growth of the *ispA*[+] strain was rescued in the presence of MC (Fig. 6h). Thus, MC appears also to prevent ROS-mediated toxicity when lipid II synthesis is completely inhibited, leading to growth in the L-form state, under osmoprotective conditions.

We also induced L-forms on isotonic NA plates with osmoprotective sucrose but without added $Mg^{2+}$ to test for a possible role of the $Mg^{2+}$ during L-form growth. In this case, no L-form growth occurred even in the presence of an *ispA* mutation, nor in wild-type cells with MC (Fig. 6h, NA + Sucrose on right panels). Thus, $Mg^{2+}$ has a critical role to support L-form growth but via a different mechanism from that of MC.

## Discussion

Previous work showed that treatment of *B. subtilis* cells with MC substantially reduces cellular iron content and rescues the growth of both *mbl* and *mreB* mutants[40]. Here, our results have established a mechanism for the protective effects of MC, involving the sequestration of iron outside the cell, restricting its intracellular availability.

The importance of iron homeostasis for β-lactam tolerance has been reported for several bacteria[34,36,73]. In this work, we have shown that iron chelation by MC is able to protect, not only cells deficient in the actin-like proteins Mbl and MreB, but also various other cell wall-deficient mutants, or cell wall-free L-forms that can be induced by treatment with antibiotics, from cell death. Taken together, these findings imply that this pathway to cell death is normally dependent on intracellular iron availability. In contrast, MC did not work to rescue the growth of a mutant lacking the alternate PG synthetic system based on aPBPs[40]. Therefore, the iron-dependent toxicity effect seems specifically to be induced when the Rod-based cell elongation system is affected. Given that the aPBP system uses the same PG precursors as the Rod system, it seems that the inactivation of aPBP activity must have a lesser impact on the metabolic re-routing that would otherwise lead to ROS toxicity. Here our data imply that the aPBP system can operate at lower PG precursor levels than can the Rod system. This is consistent with our previous finding that the cell wall expansion that occurs during the walled cell to L-form transition driven by inhibition of PG precursor synthesis depends on the aPBP system[22]. Our previous work also showed that the activity of autolytic enzymes is coupled to aPBP activity during the generation of L-forms[22]. Taken together, these findings suggest that a major role for the (non-essential) aPBP system is as a fail-safe system that protects the cells from catastrophic cell wall failure when synthesis or utilization of PG precursors is perturbed.

We went on to dissect the steps from perturbation of cell wall synthesis to cell death (Fig. 7). In line with previous findings for *B. subtilis*[57–61] and several other organisms[74,75], we found that UDP-GlcNAc is a key metabolite in the process. We showed that perturbations increasing or decreasing UDP-GlcNAc control whether or not cell death occurs, and that the death effect is dependent on excess NADH production from central carbon metabolism and RC activity. This is supported by a recent metabolomics study that the β-lactam mecillinam results in increased cellular levels of UDP-GlcNAc, glycolytic intermediates and $NAD^+$ in *E. coli*[32]. Importantly, we showed here that intracellular levels of UDP-GlcNAc are reduced in the presence of excess $Mg^{2+}$ (Supplementary Fig. 4). Elucidating how $Mg^{2+}$ influences UDP-GlcNAc levels, thereby preventing cell death upon perturbation of cell wall synthesis, is an important challenge for future studies.

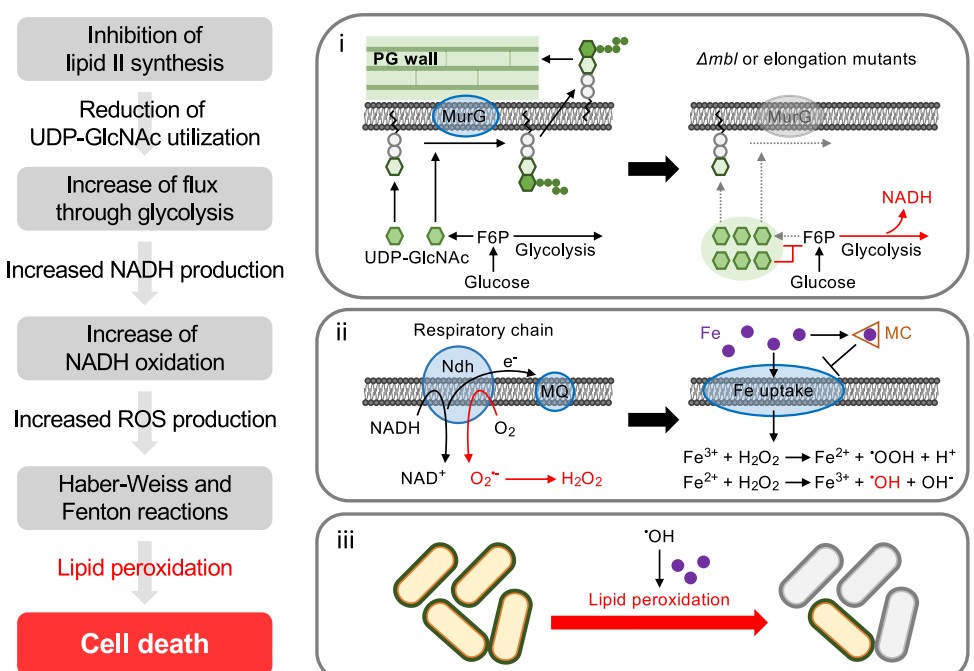

**Fig. 7 | Critical role for metabolic perturbations and ion homeostasis in the *mbl* lethality.** Schematic representation of key cellular functions that link metabolic perturbations in *mbl* mutants to bacterial cell death, and the critical connections between iron homeostasis and the killing activity. Disruption of an *mbl* gene or other elongation mutants infuences lipid II synthesis, plausibly by affecting the membrane localization or synthesis of MurG. This could cause the accumulation of UDP-GlcNAc, by preventing its utilization for lipid II synthesis (i). The increased intracellular concentration of UDP-GlcNAc is of critical importance in the down-stream metabolic shifts. When the intracellular concentration of UDP-GlcNAc is high, GlmS activity is reduced, resulting in the rerouting of glucose metabolism towards glycolysis and increased NADH production (i). Oxidation of the excess NADH in the RC pathway increases ROS generation (ii), which drives the toxic effect of the *mbl* mutation in the presence of iron (iii). The sequestration of extracellular iron in the presence of MC could reduce the labile iron pool in the cytosol acts to catalyse the production of deleterious hydroxyl radical via the Fenton/Haber-Weiss reaction cycle under oxidative stress conditions, and thereby formation of hydroxyl radical through interaction with iron is prevented (ii). This, in turn, protects the *mbl* mutant cells from toxic oxidative damage.

Another intriguing question is why perturbation of cell wall synthesis has such a severe effect, compared to other metabolic disruptions, such as abrupt changes in nutrition. However, the Gram-positive cell wall constitutes a major proportion of the mass of the cell (about 10–20%), so perhaps blocking this pathway has a particularly severe effect on the flux pathways leading to ROS.

When intracellular iron content is in excess, ROS production via cellular respiration stimulates the Fenton/Haber-Weiss reaction cycle, which catalyses the production of deleterious hydroxyl radical[67,72]. We showed that LPO, which is promoted by hydroxyl radical, also in the presence of iron[44], is the main cause of death in cell wall mutants and that this is probably responsible for the phase pale (lytic) appearance. It is commonly assumed that the gross morpho-logical changes that often accompany treatment with cell wall tar-geting antibiotics[22,76,77], or mutations perturbing cell wall synthesis (typically twisting and or bulging)[9,78,79], are the cause of subsequent lysis (phase pale appearance) and cell death. However, our studies with MC demonstrated a separation of the morphological changes and lysis, showing that there are at least two distinct pathways potentially leading to cell death upon antibiotic treatment, and support the idea that ROS can play a major role in cell killing. It also suggests that LPO, catalysed by redox-active iron, is a critical impe-diment to the growth of L-form bacteria.

In conclusion, our results highlight key enzymatic reactions that link the metabolic perturbations that occur upon cell wall inhibition to bacterial cell death, as well as critical connections between iron homeostasis and specific aspects of the killing activity. Further improvements in our understanding of the physiological changes occurring upon antibiotic treatment should prompt new strategies for antibacterial therapies.

## Methods
### Bacterial strains and growth conditions
The bacterial strains in this study are listed in Supplementary Table 1. DNA manipulations and transformations were carried out using stan-dard methods. Nutrient agar and broth (Oxoid) were used as complex rich media for bacterial growth. For iron limiting conditions, Spizizen minimal medium (SMM)[46], (0.2% (w/v) ammonium sulphate, 1.4% dipotassium phosphate, 0.6% potassium dihydrogen phosphate and 0.1% sodium citrate dihydrate), was used with supplementing 0.5% glucose, 0.02 mg/ml tryptophan, 0.02% casamino acids and 4 mM magnesium sulphate. L-form growth was induced on isotonic NA plates, composed of 2x magnesium-sucrose-maleic acid (MSM) pH7 (40 mM magnesium chloride, 1 M sucrose, and 40 mM maleic acid) mixed 1:1 with 2x NA, at 30 °C. 1 μg/ml FtsZ inhibitor, 8J[80], was added to L-form medium to prevent the growth of walled cells when required. For selections of *B. subtilis* mutants, antibiotics were added to media at the following concentrations: 1 μg/ml erythromycin, 5 μg/ml chlor-amphenicol, 60 μg/ml spectinomycin or 2.5 μg/ml kanamycin. The concentration of kanamycin was increased to 10 μg/ml in the presence of added $Mg^{2+}$. IPTG, xylose and/or $MgSO_4$ was supplemented, as appropriate. MA and MC used in this study were purified and/or syn-thesised previously[40].

### Construction of IPTG-inducible *glmU*, *murG* operon and *pdhA*, and disruption mutants of *sdhA* and *mdh*
To construct the IPTG-inducible *glmU* or *murG* mutant, the first 200–300 bp of the *glmU*, *murG* or *pdhA* gene containing Shine-Dalgarno sequence was amplified by PCR from genomic DNA of the *B. subtilis* strain 168CA, then introduced into the plasmid pMutin4[81], creating pM4-$P_{spac}$-*glmU* and pM4-$P_{spac}$-*murG*. The resulting plasmids

were introduced into the *B. subtilis* strain 168CA to generate YK1538, YK1540 and YK2732 (Supplementary Table 1), respectively. The sequences of primers used for the strain construction are listed in Supplementary Table 2.

To disrupt *sdhA* or *mdh*, the first 200–300 bp of the *sdhA* or *mdh* gene was amplified by PCR from genomic DNA of the *B. subtilis* strain 168CA, then introduced into the plasmid pMutin4[81], creating pM4-Δ*sdhA* and pM4- Δ*mdh*. The resulting plasmids were introduced into the *B. subtilis* strain 168CA to generate YK2718 and YK2720 (Supplementary Table 1), respectively. The sequences of primers are listed in Supplementary Table 2.

### Construction of xylose-inducible *murB* and *murG* gene overexpression

To construct the xylose-inducible *murB* or *murG* mutants, full length of the *murB* or *murG* gene containing Shine-Dalgarno sequence was amplified by PCR, with MK537-MK538 or MK533-MK534 primer pairs, from genomic DNA of the *B. subtilis* strain 168CA, then assembled with the plasmid pSG1728[82], creating DNA fragments containing $P_{xyl}$-*murB* and $P_{xy}$-*murG* using MK117-MK536 and MK539-MK116 primers (for *murB*), or MK117-MK532 and MK535-MK116 primers (for *murG*). The DNA fragments were introduced into *amyE* locus of an *mbl* mutant (YK2638) to generate YK2700 or YK2701 (Supplementary Table 1), respectively. The sequences of primers are listed in Supplementary Table 2.

### Microscopy and image analysis

For snapshot live cell imaging, walled cells were mounted on microscope slides covered with a thin film of 1.2% agarose in water. L-forms were mounted on microscope slides with isotonic NB containing MSM. All microscopy experiments were conducted using a Nikon Ti microscope equipped with a Nikon CFI Plan Apo DM Lambda x100 oil objective and a Photometrics Prime camera, using MetaMorph software (version 7.7, Molecular Devices). Images were analysed and processed using FIJI (version 2.9.0/1.53t, https://imagej.net/Fiji).

### Measurement of intracellular UDP-GlcNAc levels

Overnight cultures of *B. subtilis* strains, 168CA (wild-type) and YK2638 (Δ*mbl*), in LB (Luria-Bertani) liquid medium (with 10 mM $Mg^{2+}$ for Δ*mbl*) were diluted into 10 ml fresh LB with or without 10 mM $Mg^{2+}$ or 10 μg/ml MC. The cultures were incubated at 37 °C to $OD_{600nm} = 0.5$. The cells were harvested from 7 ml of culture by centrifugation and washed by resuspending the pellet in 1 ml of deionized water and subsequent centrifugation. The pellet was resuspended in 75 μl of 5% trichloroacetic acid and kept at room temperature for 20 min at 500 rpm shaking. The mixture was then centrifuged for 10 min at 15871 g. The supernatant was neutralized by the addition of 11.25 μl of KOH (2.5 M)/$K_2HPO_4$ (1.5 M) solution for LC-MS analysis. An Agilent Triple Quad 6460 linked to an Agilent LC 1290 Infinity with a CarboPac PA1 2 x 250 mm column (Thermo Fisher) was used for UDP-GlcNAc analysis. Mobile phases consisted of water (A) and 1 M ammonium acetate (B). A gradient of 25 to 30% of B over 37 min was employed at the fellow rate of 0.2 ml/min. Multiple reaction monitoring (MRM) technique with the following parameters was used for the quantitation of UDP-GlcNAc: negative polarity; precursor ion 606.07 Da; product ions 385.0 and 158.9 Da; dwell time 200 ms; collision energy for 385.0 Da was 29 V and for 158.0 was 49 V; fragmentation voltage for both product ions was 190 V and accelerating voltage was 4 V. Product ion 385.0 Da was used for quantitation. Source parameters were as follows: gas temperature 350 °C; gas fellow 12 l/min; nebulizer gas pressure 35 psi; sheath gas temperature 400 °C; sheath gas fellow 12 l/min; capillary voltage 4500 V; and nozzle voltage 500 V. Agilent MassHunter Quantitative Analysis 10.2 software was used to determine the concentration of

UDP-GlcNAc on samples based on the standard curve generated by analysis of different concentration of UDP-GlcNAc (Sigma-Aldrich) on same LC-MS condition.

### SOD activity in *B. subtilis* lysates

Overnight cultures of *B. subtilis* strains in 10 ml of PAB medium (Difco Antibiotic Medium 3) with 20 mM $MgSO_4$ ($Mg^{2+}$) were washed twice with PAB, and diluted (1:1000) into PAB with 20 mM $Mg^{2+}$ or 10 μg/ml MC. The cultures were incubated at 37 °C to $OD_{600nm} = 0.6$. The cells were harvested by centrifugation and washed with 20 mM HEPES buffer (pH 7.4), followed by a wash with 20 mM EDTA to remove surface-associated metals, then finally washed twice in PBS to remove trace EDTA.

Soluble cell extracts were prepared for SOD activity analysis by suspending cell pellets in 100 μl lysis buffer (20 mM Tris, pH 6.8, 0.1 mg/ml lysozyme, 10 μg/ml DNase, complete EDTA-free protease inhibitor tablet (Roche)), followed by incubation on ice for 1 h and subsequent sonication for 10 s at 4 °C. Soluble extracts were separated by centrifugation and total protein concentration was determined by Bradford assay. SOD activity was assessed quantitatively using an adjusted riboflavin-nitroblue tetrazolium (NBT) liquid assay method[83]. The assay was performed in 96-well plates where 20 μl of the soluble extract was mixed with 180 μl assay solution (10 mM methionine, 1.4 μM riboflavin, 66 μM NBT, and 10 μM EDTA in 50 mM potassium phosphate buffer, pH 7.8), immediately prior to 20 min incubation on a white light box followed by immediate measurement of absorbance at 560 nm using an Agilent BioTek ELx800 plate reader. Serial 2-fold dilutions of tested samples were assayed to identify sample concentrations within the linear range of the assay close to 1 U relative to a standard curve obtained using commercial Bovine SOD standard (S5639, Sigma). The fit of the linear range of the inhibition curve was generated using GraphPadPrism software (version 9.3.1) using a standard linear regression model. To calculate specific SOD activities, the assay was performed in triplicate using samples diluted to the concentrations within the assay's linear range. Specific activity was expressed in terms of enzyme units per mg total protein in the soluble extracts ($U.mg^{-1}$).

### Detection of ROS

*B. subtilis* strains were cultured in NB at 37 °C. To detect ROS (superoxide and hydroxyl radical), 1 ml of the cultures were incubated with 5 μM CellROX Green (Invitrogen) for 30 min at 37 °. The cells were harvested by centrifugation and washed three times with fresh NB before being used for microscopic analysis. CellROX Green is a proprietary oxidation-sensitive dye whose fluorescence quantum yield at 500–550 nm after excitation at 488 nm increases dramatically on oxidation in the presence of dsDNA[66].

### Detection of lipid peroxidation

Lipid peroxidation was detected using a fluorescent probe ($C_{11}$-BODIPY$^{581/591}$; Molecular Probes) as described previously[24]. *B. subtilis* strains were cultured in NB at 37 °C. To detect Lipid peroxidation, 1 ml of the cultures were incubated with 5 μM $C_{11}$-BODIPY$^{581/591}$ for 60 min at 37°. The cells were harvested by centrifugation and washed three times with fresh NB before being used for microscopic analysis.

### Reporting summary

Further information on research design is available in the Nature Portfolio Reporting Summary linked to this article.

## Data availability

Source data are provided with this paper. All data that generated in this study are provided in the Supplementary Information and Source Data file. Source data are provided with this paper.

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

## Acknowledgements

We thank Richard Daniel for the use of the *B. subtilis* strains, and the University of Sydney Mass Spectrometry Facility for access to instruments. This work was funded by grants from a European Research Council Advanced award (670980) to JE, a Wellcome Investigator Award (209500) to JE, an ARC Laureate Fellowship (FL210100071) to JE, and a BBSRC DTP studentship to KJW.

## Author contributions

Y.K. and J.E. designed concepts. Y.K. designed and performed most experiments. M.K. contributed for constructions of B. subtilis strains. E.S.M. and B.K. performed the SOD assay. Y.D. purified Mirubactin A and performed UDP-GlcNAc measurements. Y.K., K.J.W. and J.E. analysed data with contributions from all other authors. Y.K. prepared figures, wrote the original draft of the manuscript with contributions from all other authors. K.J.W. and J.E. reviewed the manuscript and edited. The project was managed by J.E.

## Competing interests

The authors declare no competing interests.
