## [Peer Review File · Nature Communications]

On the mechanisms of lysis triggered by perturbations of bacterial cell wall biosynthesisReviewer #1 (Remarks to the Author):

The study entitled "On the mechanisms of lysis triggered by perturbations of bacterial cell wall biosynthesis" by Kawai et. al., attempts to elucidate the ameliorative effect of the iron chelating siderophore, Mirubactin C, on cell wall perturbations and decouples iron-mediated cell lysis from iron-independent changes in cell morphology. While the usage of several genetic mutants within the well-conserved cell wall synthesis pathways (Rod and aPBP) to delineate the mechanism of action and metabolic underpinnings is impressive, direct experimental measurement of the downstream effects such as ROS, glycolytic intermediates, lipid peroxidation etc. is lacking. Following comments can be addressed to make the study more rigorous:

1. The authors claim that addition of Mirubactin A ameliorates $\Delta fecC$ growth deficiency however it is only a marginal recovery. Would increasing MA concentration lead to better recovery in case of $\Delta fecC$?
2. Is the steady state expression of the mur and glm genes/proteins affected in Δmbl compared to WT?
3. The rescue of Mg^{2+} via a distinct iron-independent mechanism to MC, given their overlapping effects on the Rod pathway, is intriguing and should be explored further.
4. While the authors have used a multitude of mutants to link death in Δmbl to ROS generated by cellular respiration (RC), the absence of data that directly measures ROS levels is concerning. ROS levels need to be measured for WT and Δmbl to see if there is any difference to begin with. While the RC mutants indicate perturbation in redox balance, the kind of redox dysregulation required to bring about cell death despite the presence of bacillithiol (the major redox buffer) and antioxidant enzymes, has to be drastic. This is important to rule out any inflation in the role of ROS in the observed phenotype. Perturbation of central carbon metabolism affects every arm of physiology and while redox metabolism plays an important role, it's contribution in the observed phenotype needs to be clarified further. This is important as it appears that Mg^{2+} confers protection without affecting redox balance.

Major suggestions to consider:

- a. Measurement of basal ROS in WT and Δmbl grown in NB and SMM.
- b. Measurement of ROS level in Δmbl with and without overexpression of MurG. This is important as it would prevent over-accumulation of UDP-GlcNAc and relieve the pressure on RC thereby reducing lethality via dysregulated ROS production.
- c. Measurement of UDP-GlcNAc under the different conditions as one of the major claims is increased toxicity with increased UDP-GlcNAc levels.
- d. Inclusion of a ROS scavenger like thiourea in the experiments and compare effects in the presence/ absence of MC and Mg^{2+} would strengthen the data.
- e. In addition to the mutants' data, it would be worthwhile to check NADH/NAD⁺ ratio in the conditions tested.
- f. Direct reduction of ROS by MC is critical and needs to be shown in case of L-form rescue experiment.
- g. As shown in Fig1, phase paling is observed in Δmbl grown in NB. Does this experimentally correlate with increased lipid peroxidation in the mutant?
- h. Lines 377-380 – Without actual experimental determination of superoxide or hydroxyl radicals, this statement is speculative.

Minor suggestions: The article could benefit from improving the language in certain areas to make it more scientific (Ref Lines 277-278)

As it stands, the methodology to understand the role of iron-homeostasis in ROS-mediated cell death with the use of an extracellular iron-chelator and genetic perturbations in cell wall synthesis and downstream metabolic pathways is interesting. However, further experimental data to validate the major claims and unravel novel mechanisms, distinct from iron homeostasis (which has been well studied as a mechanism of antibiotic tolerance), such as the rescue with Mg^{2+} , would considerably improve the study and make it exciting for the broad readership of Nature Communications.

Reviewer #2 (Remarks to the Author):

This study by Kawai et al. investigates the mechanism and pathway that lead to cell death during

perturbation of cell wall synthesis by either antibiotic treatment or by genetic mutations in cell wall biosynthesis genes. Different from the traditional view on the simple mode of action of cell wall targeting antibiotics for the observed bactericidal activities, more recent studies suggest that perturbation of cell wall does not necessarily cause cell death, parallel and independent mechanisms from alteration of central metabolism and ROS production caused by cell wall perturbation could be ultimately responsible for the cell death. This study uses the Gram-positive bacterium *Bacillus subtilis* as a model to comprehensively investigate such a mechanism and the pathway. They examined a chain of steps linking inhibition of cell wall biosynthesis, alteration of central metabolism, NADH production, respiratory electron transfer, ROS production, and iron homeostasis using primarily genetic approaches. This work represents a comprehensive study on the claimed cell death mechanism. This work is important. Their findings will help us better understand the mechanisms of killing by beta lactam type antibiotics. There are a couple of issues that I would like to point out and hopefully my suggestions will help the authors improve the manuscript.

First, as I point out, the authors in this study applied primarily genetic approaches, including gene mutations, conditional knockout of essential genes, dose-dependent expression of those essential genes, and simple qualitative assays on cell growth on nutrient agar. In my view, this almost exclusive genetic approach alone may not be able to provide evidence strong and convincing enough to support the claims made by the authors on metabolic rerouting, altered ROS accumulation, lipid damage, etc. Evidence is therefore preliminary in several places. For example, how dose-dependent expression of *glmU* could result in significantly different levels of metabolic sugars and cell wall precursors, how does-dependent expression of *gapA* could significantly shift the rate of glycolysis. There are several cases like the above. I am not very sure if the authors will be able to draw conclusions as stated in the subtitles based on just one or two pieces of genetic evidence.

I therefore strongly suggest that the authors diversify their approaches and provide more direct evidence. For examples, if ROS accumulation is central to the hypothesis, the authors will need to directly measure ROS levels under different conditions (e.g. in wt and different mutants, or under different dose-dependent gene expression, and/or without or with MC). I understand some of those may have been done in previous studies or by other groups, still, it is worth showing direct evidence. Another example is metabolic rerouting, a metabolic approach to assay levels of selected sugar metabolites will greatly improve the solidness of the conclusion.

Second, in connection with the above, there are quite a few places in the manuscript, the conclusions made by the authors are beyond what can be supported by genetic evidence since they have not been directly tested. For examples, the authors mentioned alteration of UDP-GlcNAc, change in NADH production, ROS levels, etc, but technically have shown none of these directly. I will suggest the authors to tone down and in some places, change how the subtitles are phrased.

The abstract can be condensed a little bit.

Lines 116-132, this paragraph at the end of introduction is supposed to provide a very brief description of the current study prior to the result section. Currently, it reads almost like the result section.

In figure 1. Maybe move the chemical structures of MC and MA from supplement to the main figure to help understand the difference between the two antibiotic derivatives.

In figure 1, given the predicted function of MC in iron utilization and the working hypothesis, it may be important to include a bacillibactin functional *B. subtilis* strain (e.g. 3610) and see if it can grow in the presence of MC, similar to what was speculated in *E. coli*.

Figure 2, since iron plays a central role in the working hypothesis, it is important to provide information about iron concentrations in NA, SMM, etc (or at least estimated concentration) not only in figure legends, but also discuss in the text.

Fig. 2D, if the reason why the *mbi* mutant grows slowly in SMM is because of the limited iron concentration in SMM, a simple experiment to verify it is to grow the *mbi* mutant in SMM but with varied concentrations of iron.

Fig. 3A-B, I am not sure if the rescue of *murGB* depletion by MC can be judged as a positive result here. It seems that MC treatment might be delaying cell death, but for how long? a CFU count may help clarify one way or the other ?

Lines 225-226, about uncoupling and growth rescue by MC, I recommend time-lapse microscopic analyses to observe the rescue by MC on the cell wall synthesis mutants. This could provide more information about whether death of those bulged cells is only delayed, for how long? Or whether those cells still manage to grow and divide, compared to without MC treatment. Addition of Mg²⁺ can be used as a positive control.

Line 245, "Thus, an increased level of PG precursor synthesis can overcome the lethal effects..." and the statement in line 249. These two statements seem premature.

PG precursors biosynthesis is a multi-step pathway involved in many other enzymes in addition to MurG, even though MurG carries the critical last step in lipid II biosynthesis. So unless the authors can demonstrate evidence or cite very solid published studies, I will suggest tone down in both statements. Plus, if cell wall synthesis (polymerization) is inhibited, shouldn't one expect increased accumulation of lipid II ? Or is a more complex feedback mechanism playing a role here ? and if so please explain.

Fig. 5, in the *mbi* mutant, the authors suggested that MurG activity is inhibited and lipid II accumulation reduced, and that the *mbi* mutation could also cause accumulation of UDP-GlcNAc. The evidence provided in this study is that overexpression of *murG* or *glmU* could partially rescue the growth defect of the *mbi* mutant. If this (gene expression and growth correlation) is the only evidence, I will be hesitant.

Fig. 5A, In the model of electron transfer, since complex III is a terminal oxidase, it is supposed to convert molecular oxygen to H₂O, instead of generating ROS, and even in complex I, ROS is probably an occasional by-product only when the electron is leaked, not delivered to MQ_o. The drawing here seems to suggest that every electron is being delivered to generate ROS.

Lines 277-279, the observation that modest expression of *glmU*, but not strong induction, can counteract the inhibitory activity of Fosfomycin is quite interesting and intriguing. According to the authors' hypothesis, this is likely due to overexpression of *glmU* leading to increased accumulation of UDP-GlcNAc (line 280), but why this increased accumulation does not happen in the vancomycin treatment ?

The use of conditional knockout construct of essential genes does not seem to be a great strategy in general, plus, in some of the inducible constructs, it also impacts other downstream genes as well.

There are different ways to alter glycolysis and central metabolism, and oxidative respiration and ROS production. If the authors would like to propose a generalized hypothesis that cell wall inhibition leads to alteration in central metabolism and thus ROS accumulation, maybe knockout or overexpression of catalase genes, growing the mutant or antibiotic treated cultures under different iron concentrations, etc, can also be considered.

Lines 271 and 283, in both places, the conclusions, if true, are very impactful, however, these conclusions are based on preliminary evidence, which is achieved by overexpressing one specific gene. This creates concern. For example, biosynthesis of UDP-GlcNAc is controlled by a multi-step pathway, whether overexpression of *glmU* is enough to significantly alter the accumulation of GlcNAc needs to be confirmed. Using biochemical approach to measure GlcNAc is not that difficult. Similarly, whether overexpression of just *gapA* can significantly increase glycolytic activities is not certain, maybe measure a few key metabolites in the glycolytic pathway to confirm increased glycolytic activities.

Minor points:

Line 74, β -Lactams...

Line 178, It is generally assumed...

Line 179, please provide a reference for this statement.

Reviewer #3 (Remarks to the Author):

In this manuscript, Kawai and colleagues study several cell-wall associated mutants and the actions of mirubactin A and C on *Bacillus subtilis* morphology and lysis. Using *mbl* mutants, which are deficient in cell elongation, the authors showed that mirubactin C could rescue their growth on rich media, whereas mirubactin A could not. Interestingly, mirubactin C did not alter the morphology of *mbl* mutants in rich media, but prevented cell lysis. Using different media supplementations and genetic constructs of peptidoglycan biosynthesis and central metabolism, the authors suggest that the lysis pathway is dependent on accumulation of UDP-GlcNAC, which leads to increased glycolytic flow, NADH production, respiration, and the associated production of reactive oxygen species. The authors note that mirubactin C helps support L-form growth, which is oxidative stress sensitive, and that it helps protect against cumene hydroperoxide by limiting iron and lipid peroxidation. Conceivably, these data can be extrapolated to additional perturbations that lead to UDP-GlcNAC accumulation and produce cell lysis. Several comments and suggestions are provided below.

1. It would be supportive to use other means to limit iron beyond mirubactin C. Citrate was used in one experiment, but other siderophores that cannot be utilized by *B. subtilis* would help establish the generality of results to iron. Conceivably reducing the available iron by using Chelex resin would phenocopy mirubactin C. The limited iron media does not appear to have a step to remove trace or control for iron contamination from media components and water.
2. It was not clear whether the mirubactin A and mirubactin C preparations added to cultures already contained iron or ions complexed with them.
3. Anaerobic experiments, perhaps with exogenous electron acceptors that are not oxygen, could be supportive of the authors conclusions about reactive oxygen species, since they conceivably arise from inadvertent transfer of electrons to dissolved oxygen.
4. Analogously, over-expression of superoxide dismutase and catalase should turn superoxide and hydrogen peroxide into water. Such experiments with the appropriate controls (perhaps catalytically-dead versions) would also provide greater support for reactive oxygen species as the crux of cell lysis from cell wall inhibition. Measuring SOD activity in the *mbl* mutant compared to wild-type is not the same as these experiments.
5. It remains to be seen whether the proposed pathway is present in other bacteria, as suggested in Figure 7. For instance, do the authors see similar results when inhibiting comparable systems in *E. coli* and using mirubactin C. It seems that the pathway and impact of mirubactin C would be translatable to other organisms.
6. Line 157: Should be "Figure 1C"
7. Figures 3A and 3B are devoid of error bars/error estimates. Overall, much of the data is pictures of plates, which at best is qualitative, and more quantitative data would have been well received.

REVIEWER COMMENTS

Reviewer #1 (Remarks to the Author):

The study entitled “On the mechanisms of lysis triggered by perturbations of bacterial cell wall biosynthesis” by Kawai et. al., attempts to elucidate the ameliorative effect of the iron chelating siderophore, Mirubactin C, on cell wall perturbations and decouples iron-mediated cell lysis from iron-independent changes in cell morphology. While the usage of several genetic mutants within the well-conserved cell wall synthesis pathways (Rod and aPBP) to delineate the mechanism of action and metabolic underpinnings is impressive, direct experimental measurement of the downstream effects such as ROS, glycolytic intermediates, lipid peroxidation etc. is lacking. Following comments can be addressed to make the study more rigorous:

>>> We hope that the extensive new results involving direct measurements included in the revised manuscript will satisfy the reviewer.

1. The authors claim that addition of Mirubactin A ameliorates $\Delta fecC$ growth deficiency however it is only a marginal recovery. Would increasing MA concentration lead to better recovery in case of $\Delta fecC$?

>>> Although addition of MA can rescue the growth of a $\Delta fecC$ strain on SMM plates, increasing mirubactin A concentration (>40 $\mu\text{g/ml}$) leads to growth inhibition (new Supplementary Fig. 1c). High MA concentrations also inhibit the growth of wild-type and $\Delta feuA$ strains on SMM plates (new Supplementary Fig. 1b). Since MA is known to spontaneously decompose into MC (Kishimoto et al., 2014, 2015), the growth inhibitory effect is likely the consequence of an increased MC concentration under the conditions used. We added these new results in revised Supplementary Figure 1 (lines 151-154).

2. Is the steady state expression of the mur and glm genes/proteins affected in Δmbl compared to WT?

>>> Unfortunately, antibodies for Mur and Glm proteins are not available to us at present, so it would be difficult to measure protein levels directly. Nevertheless, our results using the IPTG-inducible *Pspac* promoter indicate that levels of *murGB* and *glmU* gene expression that are toxic in a Δmbl background do not affect growth in a wild-type background (Figs. 3d and 4b). This suggests that the stability or activities of the proteins are likely affected by the Δmbl mutation.

3. The rescue of Mg^{2+} via a distinct iron-independent mechanism to MC, given their overlapping effects on the Rod pathway, is intriguing and should be explored further.

>>> Our new results show that both added Mg^{2+} and MC can work to reduce the increased ROS generation and lipid peroxidation of the Δmbl mutant (new Fig. 6, lines 361-387). Since the increasing SOD activity of the Δmbl mutant was reduced in the presence of added Mg^{2+} (Supplementary Fig. 6), Mg^{2+} seems to work by preventing excess superoxide production. Given that Mg^{2+} can also ameliorate the morphological defects of the Δmbl strain, it could act upstream on cell wall synthesis to compensate for the downstream metabolic problem.

Our new result also showed that Mg^{2+} works to reduce intracellular UDP-GlcNAc levels (new Supplementary Figs. 4b and c, lines 297-303), consistent with the toxic effect of the Δmbl mutation being stimulated by UDP-GlcNAc synthesis and its rescue by Mg^{2+} . However, the literature suggests that Mg^{2+} has multiple effects on cell wall homeostasis, so further work on the mechanism probably goes beyond the scope of the present paper.

In addition, we confirmed the growth rescue effect by Mg^{2+} and MC in a strain lacking the *mreBCD* genes, which are essential key factors for cell elongation (new Supplementary Fig. 2c, lines 218-220).

We have added these new results in revised manuscript (see also below).

4. While the authors have used a multitude of mutants to link death in $\Delta mb1$ to ROS generated by cellular respiration (RC), the absence of data that directly measures ROS levels is concerning. ROS levels need to be measured for WT and $\Delta mb1$ to see if there is any difference to begin with. While the RC mutants indicate perturbation in redox balance, the kind of redox dysregulation required to bring about cell death despite the presence of bacillithiol (the major redox buffer) and antioxidant enzymes, has to be drastic. This is important to rule out any inflation in the role of ROS in the observed phenotype. Perturbation of central carbon metabolism affects every arm of physiology and while redox metabolism plays an important role, its contribution in the observed phenotype needs to be clarified further. This is important as it appears that Mg^{2+} confers protection without affecting redox balance.

>>> We carried out measurements of ROS, as requested, and the results confirmed the higher levels of ROS in the $\Delta mb1$ mutant and its reduction under various conditions that rescue the growth of the mutant (e.g. Mg^{2+} , MC, *murG* upregulation, etc.). We added these new results to the revised manuscript in new Figure 6 (lines 361-387). We thank the reviewer for this suggestion, as we feel that the new data have helped to improve the manuscript.

Major suggestions to consider:

a. Measurement of basal ROS in WT and $\Delta mb1$ grown in NB and SMM.

>>> We confirmed that ROS production in the $\Delta mb1$ strain was higher than that of the wild-type in NB, but not in SMM (new Figs. 6a and b).

b. Measurement of ROS level in $\Delta mb1$ with and without overexpression of MurG. This is important as it would prevent over-accumulation of UDP-GlcNAc and relieve the pressure on RC thereby reducing lethality via dysregulated ROS production.

>>> We confirmed that either *murG* overexpression or *glmU* downregulation can work to reduce ROS production in a $\Delta mb1$ mutant (new Fig. 6b).

c. Measurement of UDP-GlcNAc under the different conditions as one of the major claims is increased toxicity with increased UDP-GlcNAc levels.

As mentioned above, we measured intracellular UDP-GlcNAc concentrations in wild-type cells, and the results showed the substantial reduction in the presence of Mg^{2+} but not with MC (new Supplementary Fig. 4b). We tried to repeat this in $\Delta mb1$ mutant cells. Although it was technically difficult to perform accurate quantitative measurements due to the phase pale or lytic phenotype of the mutant cells in standard complex liquid media (without added either Mg^{2+} or MC), we confirmed the significant reduction of UDP-GlcNAc levels in the presence of Mg^{2+} but not MC (new Supplementary Fig. 4c).

This is consistent with the idea that $\Delta mb1$ mutant is not able to tolerate higher levels of UDP-GlcNAc synthesis, and supports the possibility that Mg^{2+} works to compensate for upstream metabolic perturbation by reducing intracellular levels of UDP-GlcNAc, and thereby can avoid the toxicity. On the other hand, MC acts downstream to prevent the formation of hydroxyl radical and lipid peroxidation by restricting iron bioavailability.

We added these new results to the revised manuscript in new Supplementary Figure 4 (lines 297-303) and reworded the text according to the results.

d. Inclusion of a ROS scavenger like thiourea in the experiments and compare effects in the presence/ absence of MC and Mg²⁺ would strengthen the data.

>>> We tested this but the growth of *B. subtilis* was inhibited in the presence of thiourea.

We have previously reported that bacillithiol synthesis becomes essential for cell viability upon repression of lipid II synthesis (i.e. during the switch from walled cells to L-forms) under glycolytic conditions, but not under gluconeogenic conditions (Kawai et al., 2015, *Current biology*; 2019, *Nature Microbiology*). Our previous work also showed that intracellular bacillithiol levels are significantly reduced in wild-type cells during cell wall inhibition, but this does not occur when ROS production is reduced by a mutation in the respiratory chain. Unfortunately, bacillithiol is not commercially available for use at this time.

e. In addition to the mutants' data, it would be worthwhile to check NADH/NAD⁺ ratio in the conditions tested.

>>> Intracellular redox changes upon cell wall inhibition have been well known as a general phenomenon and have been hypothesized as a crucial contributor to cell death (e.g. Kohanski et al., 2007, *Cell*; Lobritz et al., 2022, *Cell chemical biology*).

In this paper, we show that multiple NADH generators (encoded by *gapA*, *pdhA* and *odhA*) in central carbon metabolism and NADH dehydrogenase (*ndh*) do indeed contribute to the cell death during cell wall inhibition (Fig. 5, and Supplementary Figs. 5 and 6). In addition, our new results confirmed that the ROS production in the ΔmbI mutant was reduced either by *gapA* downregulation or Δndh mutation (new Fig. 6b, lines 361-376).

f. Direct reduction of ROS by MC is critical and needs to be shown in case of L-form rescue experiment.

>>> We confirmed that ROS is reduced by MC in the ΔmbI mutant (new Figs. 6a and b).

g. As shown in Fig1, phase paling is observed in ΔmbI grown in NB. Does this experimentally correlate with increased lipid peroxidation in the mutant?

>>> Yes, we confirmed the increase in lipid peroxidation in the ΔmbI strain and its reduction in the presence of added Mg²⁺ or MC (new Fig. 6c).

h. Lines 377-380 – Without actual experimental determination of superoxide or hydroxyl radicals, this statement is speculative.

>>> New experiments using CellROX Green (fluorogenic probe for measuring production of superoxide and/or hydroxyl radicals) confirmed the higher levels of ROS in the ΔmbI mutant and its reduction under various conditions (new Fig. 6b). The new results support our conclusions (lines 361-376).

Minor suggestions: The article could benefit from improving the language in certain areas to make it more scientific (Ref Lines 277-278)

>>> Reworded (lines 287-292).

As it stands, the methodology to understand the role of iron-homeostasis in ROS-mediated cell death with the use of an extracellular iron-chelator and genetic perturbations in cell wall synthesis and downstream metabolic pathways is interesting. However, further experimental data to validate the major claims and unravel novel mechanisms, distinct from iron homeostasis (which has been well studied as a mechanism of antibiotic tolerance), such as the rescue with Mg²⁺, would considerably improve the study and make it exciting for the broad readership of *Nature*

Communications.

>>> As described above, the revised version provides extensive new data to support a link between cell wall perturbation, downstream carbon metabolism, and ultimately toxic ROS generation and lipid peroxidation. We believe that our results clarify the mechanisms underlying the pathway of ROS-mediated killing, and will contribute to an updated general view of killing by cell-wall-active antibiotics.

Our new data also suggest that Mg^{2+} works upstream to compensate for a metabolic problem by reducing intracellular UDP-GlcNAc levels, whereas MC inhibits hydroxyl radical formation and lipid peroxidation by restricting iron availability.

Reviewer #2 (Remarks to the Author):

This study by Kawai et al. investigates the mechanism and pathway that lead to cell death during perturbation of cell wall synthesis by either antibiotic treatment or by genetic mutations in cell wall biosynthesis genes. Different from the traditional view on the simple mode of action of cell wall targeting antibiotics for the observed bactericidal activities, more recent studies suggest that perturbation of cell wall does not necessarily cause cell death, parallel and independent mechanisms from alteration of central metabolism and ROS production caused by cell wall perturbation could be ultimately responsible for the cell death. This study uses the Gram-positive bacterium *Bacillus subtilis* as a model to comprehensively investigate such a mechanism and the pathway. They examined a chain of steps linking inhibition of cell wall biosynthesis, alteration of central metabolism, NADH production, respiratory electron transfer, ROS production, and iron homeostasis using primarily genetic approaches. This work represents a comprehensive study on the claimed cell death mechanism. This work is important. Their findings will help us better understand the mechanisms of killing by beta lactam type antibiotics. There are a couple of issues that I would like to point out and hopefully my suggestions will help the authors improve the manuscript.

>>> We thank the referee for their positive remarks. We have added extensive new data in the revised version that provide further evidence to support the conclusions and improve the manuscript.

First, as I point out, the authors in this study applied primarily genetic approaches, including gene mutations, conditional knockout of essential genes, dose-dependent expression of those essential genes, and simple qualitative assays on cell growth on nutrient agar. In my view, this almost exclusive genetic approach alone may not be able to provide evidence strong and convincing enough to support the claims made by the authors on metabolic rerouting, altered ROS accumulation, lipid damage, etc. Evidence is therefore preliminary in several places. For example, how dose-dependent expression of *glmU* could result in significantly different levels of metabolic sugars and cell wall precursors, how dose-dependent expression of *gapA* could significantly shift the rate of glycolysis. There are several cases like the above. I am not very sure if the authors will be able to draw conclusions as stated in the subtitles based on just one or two pieces of genetic evidence. I therefore strongly suggest that the authors diversify their approaches and provide more direct evidence. For examples, if ROS accumulation is central to the hypothesis, the authors will need to directly measure ROS levels under different conditions (e.g. in wt and different mutants, or under different dose-dependent gene expression, and/or without or with MC). I understand some of those may have been done in previous studies or by other groups, still, it is worth showing direct evidence. Another example is metabolic rerouting, a metabolic approach to assay levels of selected sugar metabolites will greatly improve the solidness of the conclusion.

>>> In addition to the genetic evidence, we have now carried out further experiments to directly measure ROS levels. The results confirmed that both Mg^{2+} and MC rescue ΔmbI growth by reducing the increased ROS production and lipid peroxidation. We have also confirmed that the growth rescue effects on the ΔmbI mutant under various conditions (*murG* overexpression, *glmM* downregulation, *Andh*, etc.) correlate with reduced ROS levels. We have added these new results in new Figure 6 (lines 361-387).

Our new data also show that added Mg^{2+} reduces intracellular UDP-GlcNAc levels (new Supplementary Fig. 4, lines 297-303). This supports the hypothesis that Mg^{2+} works to compensate for metabolic perturbation by reducing UDP-GlcNAc levels, thereby avoiding downstream toxic ROS production.

Since recent papers, including our own study, had reported an increased carbon flux through glycolysis during cell wall inhibition in *E. coli* (Lobritz et al., 2022, Cell Chemical Biology) and *B. subtilis* (Kawai et al 2019, Nature Microbiology), we did not revisit this effect here. Instead, we tested the effects of activating gluconeogenesis on growth and ROS levels in the ΔmbI strain. Strikingly, activating gluconeogenesis by supplying a gluconeogenic carbon source (malate) in NB rescued ΔmbI mutant growth by reducing toxic ROS production (new Fig. 6b and new Supplementary Fig. 5c, lines 324-326 and 361-376).

Second, in connection with the above, there are quite a few places in the manuscript, the conclusions made by the authors are beyond what can be supported by genetic evidence since they have not been directly tested. For examples, the authors mentioned alteration of UDP-GlcNAc, change in NADH production, ROS levels, etc, but technically have shown none of these directly. I will suggest the authors to tone down and in some places, change how the subtitles are phrased.

>>> Language and some subtitles tempered as suggested.

We also carried out measurements of ROS (see above) and UDP-GlcNAc. We initially measured intracellular UDP-GlcNAc levels in wild-type cells with or without added Mg^{2+} or MC and found a substantial reduction with Mg^{2+} . We then attempted to repeat this in an ΔmbI mutant. However, the phase pale effect in ΔmbI mutant cells interferes with performing accurate quantitative analysis in standard NB medium (without added Mg^{2+} or MC). Nevertheless, we again observed the significant reduction of UDP-GlcNAc levels with Mg^{2+} compared with MC. We added these new results to the revised manuscript (new Supplementary Fig. 4, lines 297-303).

Our new data support the hypothesis that ΔmbI mutant does not tolerate higher levels of UDP-GlcNAc synthesis. This also reinforces that Mg^{2+} works to compensate for upstream metabolic perturbation, thereby avoiding the ROS-mediated toxicity in ΔmbI mutant cells. We reworded the text according to these new results.

The abstract can be condensed a little bit.

>>> Done.

Lines 116-132, this paragraph at the end of introduction is supposed to provide a very brief description of the current study prior to the result section. Currently, it reads almost like the result section.

>>> Reduced as suggested.

In figure 1. Maybe move the chemical structures of MC and MA from supplement to the main figure to help understand the difference between the two antibiotic derivatives.

>>> Done.

In figure 1, given the predicted function of MC in iron utilization and the working hypothesis, it may be important to include a bacillibactin functional *B. subtilis* strain (e.g. 3610) and see if it can grow in the presence of MC, similar to what was speculated in *E. coli*.

>>> We obtained a bacillibactin-functional Marburg strain (NCIB3610) and confirmed the prediction of MC resistant growth on SMM plates (new Fig. 1g, lines 165-166).

Figure 2, since iron plays a central role in the working hypothesis, it is important to provide information about iron concentrations in NA, SMM, etc (or at least estimated concentration) not only in figure legends, but also discuss in the text.

>>> The Fe content of nutrient broth/agar was previously estimated to be 0.274 mg/L (Theodore and Schade, 1965) derived from the blood extract it contains (line 136). However, the complex chemical composition of this medium means that the extent to which Fe is available and not chelated is very difficult to predict. Thus, the absolute iron concentration will probably not accurately reflect the iron availability to the microbes. Nonetheless, nutrient broth/agar is generally considered a 'universal' medium for culturing diverse microorganisms, so is not considered to be either high or low in Fe but Fe-replete.

As suggested for Reviewer 3, we reduced the available iron in NB by using Chelex resin and observed sustained growth of the Δmbl mutant, with abnormal morphologies but without the appearance of significant numbers of phase pale cells (new Fig. 2d), just as was seen for MC treatment (Fig. 2b) (lines 190-193).

On the other hand, the SMM, with no added iron, should be highly Fe-deficient (levels of Fe were below the levels of detection in our ICP analysis). Only trace Fe should be present as a minor contaminant from the chemicals added or due to trace elements leaching from the glassware. We clarified this in the revised text.

Fig. 2D, if the reason why the *mbl* mutant grows slowly in SMM is because of the limited iron concentration in SMM, a simple experiment to verify it is to grow the *mbl* mutant in SMM but with varied concentrations of iron.

>>> Addition of iron to SMM (100-200 μM) improved the growth of the Δmbl strain. However, further addition (>500 μM) resulted in precipitation. We also found that not only growth but also the cell morphology of Δmbl was ameliorated in SMM (without added iron) (new Fig. 2d). Although we confirmed the expected reduction of ROS production by Δmbl in SMM (new Fig. 6b), additional factors other than iron could also contribute to the ROS levels under these conditions. We mentioned this in the text (lines 201-205).

Fig. 3A-B, I am not sure if the rescue of murGB depletion by MC can be judged as a positive result here. It seems that MC treatment might be delaying cell death, but for how long? a CFU count may help clarify one way or the other ?

>>> Our recent papers had investigated cell viability when lipid II synthesis is inhibited (Kawai et al., 2018, Cell; 2019, Nature Microbiology). Even if ROS-mediated killing is prevented during lipid II inhibition, prolonged culture of the cells results ultimately in osmotic lysis due to the loss of cell wall integrity. However, the osmotic lysis is suppressed under osmoprotective conditions and cell wall-

deficient cells can be formed (i.e. protoplasts or L-forms). Supplementary Figure 3b shows that cell growth does not occur on standard NA plates with or without MC when lipid II synthesis is completely blocked, but growth (in the L-form state) is sustained on sucrose-NA plates when MC is added (Fig. 6h). So, MC treatment works to prevent ROS-mediated killing by *murGB* depletion but not for osmotic lysis.

Lines 225-226, about uncoupling and growth rescue by MC, I recommend time-lapse microscopic analyses to observe the rescue by MC on the cell wall synthesis mutants. This could provide more information about whether death of those bulged cells is only delayed, for how long? Or whether those cells still manage to grow and divide, compared to without MC treatment. Addition of Mg²⁺ can be used as a positive control.

>>> We had previously shown time-lapse microscopic experiments during lipid II inhibition and proposed a mechanism for the development of bulges when ROS-mediated killing is prevented (Kawai et al 2018, Cell).

The development of bulges is promoted by a dispersed mode of PG synthesis supported by the aPBP system, which can become operational when the Rod elongation system is perturbed, as shown for *rodA* repression (Fig. 2f and Supplementary Fig. 2d). They can grow and divide if PG precursors are available for cell wall expansion. However, depletion of precursors ultimately results in the formation of cell wall lesions by ongoing autolytic activity, leading to osmotic lysis on standard NA plates (Supplementary Fig. 3b) or L-form growth on osmoprotective NA plates (Fig. 6h), as described above.

We clarified this in the revised text (lines 235-237 and 418-420). However, since it had been reported previously, we felt it was of limited value to go further into this in this work.

Line 245, "Thus, an increased level of PG precursor synthesis can overcome the lethal effects..." and the statement in line 249. These two statements seem premature.

PG precursors biosynthesis is a multi-step pathway involved in many other enzymes in addition to MurG, even though MurG carries the critical last step in lipid II biosynthesis. So unless the authors can demonstrate evidence or cite very solid published studies, I will suggest tone down in both statements. Plus, if cell wall synthesis (polymerization) is inhibited, shouldn't one expect increased accumulation of lipid II? Or is a more complex feedback mechanism playing a role here? and if so please explain.

>>> Section reworded.

Fig. 5, in the *mbl* mutant, the authors suggested that MurG activity is inhibited and lipid II accumulation reduced, and that the *mbl* mutation could also cause accumulation of UDP-GlcNAc. The evidence provided in this study is that overexpression of *murG* or *glmU* could partially rescue the growth defect of the *mbl* mutant. If this (gene expression and growth correlation) is the only evidence, I will be hesitant.

>>> We have confirmed that both *murG* upregulation and *glmU* downregulation can work to sustain the growth of a Δmbl mutant by reducing the increased ROS production (new Fig. 6b, lines 361-376)).

However, in those cases, cell wall growth by the Rod system is still not fully functional as judged by morphological abnormalities (Figs. 3e and 4c). It could affect the efficiency of cell growth.

Fig. 5A, In the model of electron transfer, since complex III is a terminal oxidase, it is supposed to convert molecular oxygen to H₂O, instead of generating ROS, and even in complex I, ROS is probably an occasional by-product only when the electron is leaked, not delivered to MQ. The drawing here seems to suggest that every electron is being delivered to generate ROS.

>>> Redrawn to clarify.

Lines 277-279, the observation that modest expression of *glmU*, but not strong induction, can counteract the inhibitory activity of Fosfomycin is quite interesting and intriguing. According to the authors' hypothesis, this is likely due to overexpression of *glmU* leading to increased accumulation of UDP-GlcNAc (line 280), but why this increased accumulation does not happen in the vancomycin treatment ?

>>> Since vancomycin acts directly by binding to lipid II and perhaps sequesters lipid II, no significant negative impact may occur on de novo lipid II synthesis or UDP-GlcNAc utilization.

The use of conditional knockout construct of essential genes does not seem to be a great strategy in general, plus, in some of the inducible constructs, it also impacts other downstream genes as well. There are different ways to alter glycolysis and central metabolism, and oxidative respiration and ROS production. If the authors would like to propose a generalized hypothesis that cell wall inhibition leads to alteration in central metabolism and thus ROS accumulation, maybe knockout or overexpression of catalase genes, growing the mutant or antibiotic treated cultures under different iron concentrations, etc, can also be considered.

>>> We have used knockout mutants for the *ΔmbI* rescue phenotype (i.e. *Δndh*, *Δodh*) (Fig. 5e and Supplementary Fig. 6).

In addition, we now show that activating gluconeogenesis by supplying a gluconeogenic carbon source (malate) in NB can rescue the growth of a *ΔmbI* strain by reducing toxic ROS production (new Fig. 6b and new Supplementary Fig. 4c, lines 324-326 and 361-376).

We have also tested effects of *kata* (major catalase) overexpression on *ΔmbI* growth and ROS accumulation, but no significant effects were observed (please also see the reviewer 3 below).

Lines 271 and 283, in both places, the conclusions, if true, are very impactful, however, these conclusions are based on preliminary evidence, which is achieved by overexpressing one specific gene. This creates concern. For example, biosynthesis of UDP-GlcNAc is controlled by a multi-step pathway, whether overexpression of *glmU* is enough to significantly alter the accumulation of GlcNAc needs to be confirmed. Using biochemical approach to measure GlcNAc is not that difficult. Similarly, whether overexpression of just *gapA* can significantly increase glycolytic activities is not certain, maybe measure a few key metabolites in the glycolytic pathway to confirm increased glycolytic activities.

>>> Our result shows that reducing *glmU* expression works to increase fosfomycin resistance, but restoring its expression returns fosfomycin sensitivity to the normal level (Fig. 4e). We do not infer from this that *glmU* overexpression increases the accumulation of UDP-GlcNAc over that of the wild-type. We clarified this in the revised manuscript (lines 287-292).

Similarly, we agree that whether overexpression of *gapA* can significantly increase glycolytic activities is not certain. However, *gapA* repression should reduce the flux through glycolysis. Many other glycolytic enzymes are bifunctional, acting in both glycolysis and gluconeogenesis, except for *GapA*, which specifically and essentially acts on glycolysis. It was for this reason that we selected *gapA* to test the effect on glycolysis.

In addition, as mentioned above, our previous carbon flux analysis showed an increased flux through glycolysis during cell wall inhibition (Kawai et al 2019, Nature Microbiology). Furthermore, we now show that activating gluconeogenesis can rescue the growth of the *Δmbl* mutant (new Supplementary Fig. 5b, lines 324-326)) and reduce the increased ROS production (new Fig. 6b, lines 361-376)).

Minor points:

Line 74, β-Lactams...

Line 178, It is generally assumed....

Line 179, please provide a reference for this statement.

>>> All corrected.

Reviewer #3 (Remarks to the Author):

In this manuscript, Kawai and colleagues study several cell-wall associated mutants and the actions of mirubactin A and C on *Bacillus subtilis* morphology and lysis. Using *mbl* mutants, which are deficient in cell elongation, the authors showed that mirubactin C could rescue their growth on rich media, whereas mirubactin A could not. Interestingly, mirubactin C did not alter the morphology of *mbl* mutants in rich media, but prevented cell lysis. Using different media supplementations and genetic constructs of peptidoglycan biosynthesis and central metabolism, the authors suggest that the lysis pathway is dependent on accumulation of UDP-GlcNAC, which leads to increased glycolytic flow, NADH production, respiration, and the associated production of reactive oxygen species. The authors note that mirubactin C helps support L-form growth, which is oxidative stress sensitive, and that it helps protect against cumene hydroperoxide by limiting iron and lipid peroxidation. Conceivably, these data can be extrapolated to additional perturbations that lead to UDP-GlcNAC accumulation and produce cell lysis. Several comments and suggestions are provided below.

1. It would be supportive to use other means to limit iron beyond mirubactin C. Citrate was used in one experiment, but other siderophores that cannot be utilized by *B. subtilis* would help establish the generality of results to iron. Conceivably reducing the available iron by using Chelex resin would phenocopy mirubactin C. The limited iron media does not appear to have a step to remove trace or control for iron contamination from media components and water.

>>> We showed that reducing the available iron in NB through pre-treatment with 1% Chelex resin prevents the appearance of significant numbers of phase pale cells (new Fig. 2d), as seen for MC (Fig. 2b). This new result was added to the revised manuscript (lines 190-193).

2. It was not clear whether the mirubactin A and mirubactin C preparations added to cultures already contained iron or ions complexed with them.

>>> We used the synthetic mirubactin C that is an iron-free form. The mirubactin A is a purified natural compound and also in the iron-free form (Kepplinger et al., 2022, Front Microbiol). We have mentioned this in the text (lines 137 and 147).

3. Anaerobic experiments, perhaps with exogenous electron acceptors that are not oxygen, could be supportive of the authors conclusions about reactive oxygen species, since they conceivably arise from inadvertent transfer of electrons to dissolved oxygen.

>>> Although historically *B. subtilis* was thought to be a strict aerobe, it has been reported as able to grow anaerobically, using nitrate as a terminal electron acceptor, under certain conditions. We tried to culture *B. subtilis* wild-type and *mbl* mutant strains anaerobically on nutrient agar plates in the presence of added nitrate, but no growth was detected for either strain.

4. Analogously, over-expression of superoxide dismutase and catalase should turn superoxide and hydrogen peroxide into water. Such experiments with the appropriate controls (perhaps catalytically-dead versions) would also provide greater support for reactive oxygen species as the crux of cell lysis from cell wall inhibition. Measuring SOD activity in the *mbl* mutant compared to wild-type is not the same as these experiments.

>>> We have tested the effects of ectopic overexpression of *sodA* (major superoxide dismutase) and *katA* (major catalase) on the growth of Δmbl , but no significant effects were observed. Since SodA is known to be a highly abundant protein (Eymann et al., 2004, Proteomics), this is perhaps not surprising. Although *katA* gene expression is dependent on intracellular ROS levels, we observed significant *katA* promoter activity in wild-type cells under our growth conditions. We also observed upregulation of the *katA* promoter in the Δmbl mutant.

However, since SOD and Catalase are very highly active enzymes that require metal cofactors, it's possible that increasing their expression alone is not enough to significantly dampen the production of ROS, which are endogenously produced as byproducts of natural processes. For example, metal delivery to one or other enzyme might be limiting, one or other enzyme may not be localised sufficiently near the ROS source to fully eliminate it, or the second, poorly characterised SOD in *B. subtilis*, SodF, may play an important role here. We will investigate this in future.

Nonetheless, we have now carried out experiments to directly measure ROS levels, and the results confirmed that both Mg^{2+} and MC reduce both the ROS production and lipid peroxidation observed in the Δmbl mutant. We have also confirmed that the growth rescue effects on the Δmbl mutant under various conditions (*murG* overexpression, *glmM* downregulation, *Andh*, etc.) correlate with reduced ROS levels. We have added these new results in Figure 6 (lines 361-387).

5. It remains to be seen whether the proposed pathway is present in other bacteria, as suggested in Figure 7. For instance, do the authors see similar results when inhibiting comparable systems in *E. coli* and using mirubactin C. It seems that the pathway and impact of mirubactin C would be translatable to other organisms.

>>> As shown in figure 1f, mirubactin C does not inhibit *E. coli* cells, since they produce an endogenous siderophore that can counteract the effect of mirubactin C.

Although the iron-mediated killing upon cell wall inhibition has been recently recognised in various bacteria, the mechanism and pathway remains poorly characterised. We plan to investigate the extent to which our proposed mechanism can be extended to other bacteria in future work.

6. Line 157: Should be "Figure 1C"

>>> Corrected.

7. Figures 3A and 3B are devoid of error bars/error estimates. Overall, much of the data is pictures of plates, which at best is qualitative, and more quantitative data would have been well received.

>>> Error bars added. As mentioned above, we have added quantitative data that support the conclusions and improve the manuscript in response to the reviewers' comments. We hope that the improvements within the revised version satisfy the reviewer.

Reviewer #1 (Remarks to the Author):

The authors have done a good job in addressing the points we raised in our original review. We recommend the revised paper for publication in Nature Communications.

Reviewer #2 (Remarks to the Author):

In this study, the authors investigated the mechanism and pathway that lead to cell death during perturbation of cell wall synthesis. This study is comprehensive, involving ideas spanning from iron availability and homeostasis, iron and metabolism-triggered ROS generation and oxidative stress, cell wall biosynthesis, metabolic rerouting/perturbation, etc. The authors try to propose a model that when treated with wall-targeting antibiotics, the ultimate cell death is also distinctly contributed by metabolic perturbation, increased ROS generation, and damages caused by ROS. The use of MC to chelate extracellular iron and thus reduce iron uptake was used in this study to probe and test the above mechanism. Overall, this study is interesting, comprehensive, and important. In the revised manuscript, the authors performed several additional experiments, as requested by reviewers, on direct measurements on ROS levels and certain metabolic sugars. These evidence significantly helps to support the authors' claims. The revisions are substantial and satisfactory in general. I don't have any major comments.

One thing I am still curious is that under normal conditions, metabolic perturbation and changes in iron availability are fairly common in the cells and they rarely become a severe issue that could cause cell death. While in the *mbl/mreB* mutant or when cells treated with cell wall-targeting antibiotics, those cells become extremely sensitive to metabolic rerouting/perturbation and iron availability. There must have been some type of strong synthetic lethality involved here. Maybe the authors could provide some discussion/comment on this.

Line 420, please fix the reference format.

REVIEWERS' COMMENTS

Reviewer #1 (Remarks to the Author):

The authors have done a good job in addressing the points we raised in our original review. We recommend the revised paper for publication in Nature Communications.

>>> We thank the referee for his/her positive message.

Reviewer #2 (Remarks to the Author):

In this study, the authors investigated the mechanism and pathway that lead to cell death during perturbation of cell wall synthesis. This study is comprehensive, involving ideas spanning from iron availability and homeostasis, iron and metabolism-triggered ROS generation and oxidative stress, cell wall biosynthesis, metabolic rerouting/perturbation, etc. The authors try to propose a model that when treated with wall-targeting antibiotics, the ultimate cell death is also distinctly contributed by metabolic perturbation, increased ROS generation, and damages caused by ROS. The use of MC to chelate extracellular iron and thus reduce iron uptake was used in this study to probe and test the above mechanism. Overall, this study is interesting, comprehensive, and important. In the revised manuscript, the authors performed several additional experiments, as requested by reviewers, on direct measurements on ROS levels and certain metabolic sugars. These evidence significantly helps to support the authors' claims. The revisions are substantial and satisfactory in general. I don't have any major comments.

>>> We appreciate the referee's positive comments on our manuscript.

One thing I am still curious is that under normal conditions, metabolic perturbation and changes in iron availability are fairly common in the cells and they rarely become a severe issue that could cause cell death. While in the *mbI/mreB* mutant or when cells treated with cell wall-targeting antibiotics, those cells become extremely sensitive to metabolic rerouting/perturbation and iron availability. There must have been some type of strong synthetic lethality involved here. Maybe the authors could provide some discussion/comment on this.

>>> We think that the extreme toxicity in the cell wall mutants is because ROS production is greater than during other metabolic perturbations. The Gram-positive wall comprises a huge proportion (~10-20%) of the total cell mass, so when the pathway is blocked it has a particularly big effect on central carbon metabolism. We've put two extra sentences in the Discussion to comment on this (lines 468-472).

Line 420, please fix the reference format.

>>> Corrected.